

# Measurement report: Influence of long-range transported dust on cirrus cloud formation over remote ocean: Case studies near Midway Island, Pacific

Huijia Shen[1,a], Zhenping Yin[2,a], Yun He[1,3,*], Longlong Wang[2], Yifan Zhan[4], Dongzhe Jing[1,3]

[1]School of Electronic Information, Wuhan University, Wuhan 430072, China
[2]School of Remote Sensing and Information Engineering, Wuhan University, Wuhan 430072, China.
[3]State Observatory for Atmospheric Remote Sensing, Wuhan 430072, China.
[4]Shanghai Institute of Satellite Engineering, Shanghai 201109, China
[a]These authors contributed equally to this work.

*Correspondence to:* Yun He (heyun@whu.edu.cn)

**Abstract.** Cirrus clouds play an essential role in regulating the global radiative balance and climate by both reflecting the incoming shortwave solar radiation and reserving the outgoing longwave radiation in the atmosphere. The cirrus-induced net radiative forcing is mainly determined by their microphysical properties, which are strongly associated with the competition between two ice-nucleating mechanisms, i.e., heterogeneous and homogeneous nucleation. However, it is still not well understood whether the long-range transoceanic dust can potentially urge heterogeneous nucleation to the initial ice formation in cirrus clouds even farther over vast remote ocean regions and the response of dominant ice-nucleating mechanism to the concentrations of available ice nucleating particles (INPs). Here we report on the influence of transpacific dust plumes on the ice formation in cirrus clouds via heterogeneous nucleation based on the combined observations of space-borne instruments, i.e., the Cloud-Aerosol Lidar with Orthogonal Polarization (CALIOP) and Cloud Profiling Radar (CPR). Two cases near Midway Island (28.21°N, 177.38°W), located in the central Pacific, are studied, in which the long-range transported dust plumes originate from intense Asian dust events. For both cases, partial cloud parcels show the typical in-cloud ice crystal number concentrations (ICNC) of <100 L$^{-1}$ for heterogeneous nucleation with a good agreement (within an order of magnitude) of in-cloud ICNC and nearby dust-related INP concentration (INPC) values, indicating that dust-related heterogeneous nucleation is dominated in ice formation. In addition, for the other parts of clouds without sufficient INP supply, homogeneous nucleation can still be dominated with ICNC values exceeding 300 L$^{-1}$. Therefore, dust events with sufficient intensity are capable of conducting long-range transport and influencing cirrus formation over remote ocean regions. This study shows that the natural supply of effective INPs to the upper troposphere, such as long-range transported dust aerosols can increase the cloud cover to reflect more solar radiation over oceanic regions and modulate the microphysical properties of cirrus clouds through different ice-nucleating regimes, both of which may further result in a cooling effect on global climate and should be well considered in climate evaluation.



## 1. Introduction

Cirrus clouds are ice clouds that widely exist in the upper troposphere and lower stratosphere when temperatures are lower than -38℃ (Heymsfield et al., 2017; Zou et al., 2020), with the occurrence ranging from 33% in the tropics to 7% in polar regions (Sassen et al., 2008). Cirrus clouds play a vital role in global climate by regulating the radiative balance of the Earth (IPCC, 2021). Cirrus clouds reflect the incoming shortwave solar radiation to space, thus resulting in a cooling effect (also called 'solar albedo effect'); while similar to the 'greenhouse effect', cirrus clouds can absorb the longwave radiation from the lower atmosphere and Earth's surface and emit them back to the lower atmosphere, causing a warming effect.

The net radiation of cirrus clouds is essentially determined by the competition between 'solar albedo' and 'greenhouse' effects (Liou, 1986; Baran, 2004), which mainly depends on the microphysical properties of ice crystals within the clouds. Ice crystals within cirrus clouds can be formed via either homogeneous nucleation (spontaneous) or heterogeneous nucleation (with the aid of ice-nucleating particles), thus causing a large uncertainty in general circulation models (Krämer et al., 2016, 2020). Cirrus clouds formed by homogeneous nucleation usually contain small ice crystals with a relatively larger ice crystal number concentration (ICNC) and tend to result in a net warming effect; in contrast, cirrus clouds formed by heterogeneous nucleation have a larger size but a lower ICNC, which may weaken the warming effect and even cause a possible transition from net warming to net cooling (DeMott et al., 2010; Lohmann and Gasparini, 2017). It is still difficult to conclude the exact impact of cirrus clouds on global climate so far (warming or cooling) (Wolf et al., 2019). Additionally, this is also the basic thought of geoengineering in which artificial seeding of ice nucleating particles (INPs) is proposed to thin cirrus clouds and further offset global climate warming (Gasparini and Lohmann, 2016; Liu and Shi, 2021).

Homogeneous nucleation usually requires a high ice saturation exceeding 140-150% (Koop et al., 2000). However, in situ measurements conducted by Dekoutsidis et al. (2023) show that the occurrence of in-cloud ice saturation of higher than 140% is unusual, with probability densities of less than 1% at cirrus cloud altitude (see figure 2 therein). Differently, heterogeneous nucleation can take place once ice saturation is just over 100% as long as effective INPs are provided. Dust aerosols as one of the most efficient INPs can be elevated to the middle and upper troposphere. Due to the large emission quantity, they are essential to the freezing process in mixed-phase and ice clouds (Villanueva et al., 2020). Numerous remote-sensing and in-situ observations have indicated that dust-induced ice production plays a crucial role in global cirrus clouds formation (Cziczo et al., 2013; Froyd et al., 2022; Ansmann et al., 2019a; He et al., 2022a, 2022b), especially at the upper-troposphere dust belt in the Northern Hemisphere (Yang et al., 2022). The dust-cirrus interactions are widely studied for terrestrial and offshore regions (Huang et al., 2006; Nee et al., 2007; Froyd et al., 2010; Sakai et al., 2014; Ansmann et al., 2019a). However, over extensive ocean regions, few cirrus cloud observations are reported and the formation of cirrus clouds has not yet been investigated (Comstock et al., 2002; Fujiwara et al., 2009; Cairo et al., 2021). It's well-accepted that remote maritime atmospheric environment is usually clean with an aerosol optical depth of less than 0.1 and is composed of marine aerosol contained within the lower troposphere (Smirnov et al., 2011). Nevertheless, the occurrence rate of cirrus clouds over the remote ocean can reach up to 0.3-0.4, even if without considering the regions from 15°S to 15°N where the tropical tropopause layer cirrus clouds associated with deep convection are frequently present (Sassen et al., 2008; Riihimaki and McFarlane, 2010). Are the formation of cirrus clouds purely attributed to homogeneous freezing mode over the ocean?

It is well known that Saharan dust from North Africa can be transported across the Atlantic to North America (Yu et al., 2021; Dai et al., 2022); besides, Asian dust plumes can undergo long-range transport to the North Pacific (Huang et al., 2008) and Northern America (Guo et al., 2017), and even full circle around the globe (Uno et al., 2009). Hence, the possible influence of transoceanic dust particles on cirrus cloud formation should be concerned. During the NASA CRYSTAL-FACE (Cirrus Regional Study of Tropical Anvils and Cirrus Layers - Florida Area Cirrus Experiment) program in July 2002, it is found that transatlantic Saharan dust can reach as far as Florida and impact ice formation in cirrus and supercooled altocumulus clouds (Saseen et al. 2003; DeMott et al., 2003). Also, Saseen et al. (2003) mentioned that it remains to be determined whether such ice formation triggered by transoceanic dust in cirrus clouds is a universal manner over the planet Earth. A similar impact on



ice formation in cirrus clouds is also observed for transpacific Asian dust over the Western United States (Sassen, 2001). He et al. (2022b) observed that transported Asian dust can influence the ice formation of cirrus clouds over central China by acting as INPs and discussed the ice-nucleating mechanism in depth with quantitative height-resolved ICNC and dust INP concentration (INPC). Further, it is also of great interest to quantificationally examine whether transoceanic dust plumes can provide high-level INPC values farther over the central Pacific and thus alter the dominant ice nucleating mechanism over the

remote ocean. In addition, these overlying high-latitude cirrus clouds comprised of larger ice crystals may have an impact on the development of underlying marine stratocumulus through both the seeder-feeder and radiative effects (Jian et al., 2022). Therefore, it is indispensable for understanding the potential dust-cirrus interactions over the remote ocean.

At cirrus altitudes, an INP can primarily generate a single ice crystal via heterogeneous nucleation since secondary ice nucleation is not involved below -38℃ (Ansmann et al., 2019a). Thus, if the closure between in-cloud ICNC and dust INPC

nearby can be realized, it can conclude that dust-induced heterogeneous nucleation dominates the ice formation within cirrus clouds (He et al., 2022b). Otherwise, homogeneous nucleation may participate in increasing ICNC far beyond dust INPC. As a result, the ICNC-INPC closure study can be used to verify the exact ice nucleation regime that occurs during the passage of long-range transported dust plumes. The dust INPC can be derived by the observations of a spaceborne lidar Cloud-Aerosol Lidar with Orthogonal Polarization (CALIOP) onboard the Cloud-Aerosol Lidar and Infrared Pathfinder Satellite Observation

(CALIPSO) (Winker et al., 2007) together with the Polarization Lidar PHOtometer Networking (POLIPHON) method (Mamouri and Ansmann, 2014, 2015, 2016, 2017, 2023). The POLIPHON method has been proven to be well performed in deriving the profiles of aerosol-type-dependent INPC, CCN, and mass concentration in many studies (Marinou et al., 2019; Ilić et al., 2021; Wieder et al., 2022; Choudhury and Tesche, 2022). The concurrent in-cloud ICNC values can be provided by the DARDAR (liDAR–raDAR) product (Sourdeval et al., 2018; Gryspeerdt et al., 2018), in which the observations of CALIOP

and spaceborne radar Cloud Profiling Radar (CPR) onboard the CloudSat satellite are combined and is proven to be reliable (Marinou et al., 2019; Krämer et al., 2020).

This study focuses on examining the potential influence of long-range transported dust particles on the formation of cirrus clouds over remote oceanic regions based on the observations of CALIOP and CPR. In figure 1, the second Modern-Era Retrospective analysis for Research and Applications (MERRA-2) annual mean dust column mass density in 2010 depicts an

evident transpacific dust belt to the north of 20°N. Therefore, two cases near Midway Island (28.21°N, 177.38°W), located in the central Pacific within this transpacific dust belt, are studied in detail. The organization of this paper is as follows. Section 2 introduces briefly the observational data and methodology first. Then, two case studies on cirrus clouds formed via heterogeneous nucleation in the vicinity of dust INPs are provided. In section 4, further discussions and conclusions are presented.

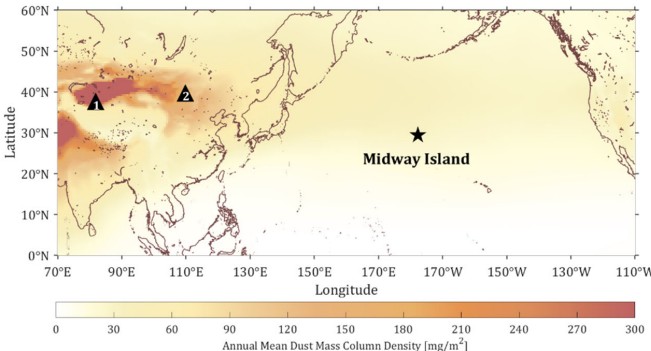

**Figure 1. The annual mean dust column mass density in 2010 provided by the MERRA-2 data. The geographical location of Midway Island (28.21°N, 177.38°W) is denoted by the black star. The East Asian dust sources, i.e., (1) Taklimakan desert and (2) Gobi, are marked with black triangles, respectively.**



## 2. Data and Methodology

### 2.1 CALIOP observational data

The spaceborne polarization lidar CALIOP carried on the CALIPSO satellite was utilized to obtain the vertical profiles of optical properties of cirrus clouds and dust layers. The CALIOP can detect the 532-nm and 1064-nm elastic backscatters together with the 532-nm volume linear depolarization ratio. The level-1 product (version 4.51) was used to provide the 532-nm total attenuated backscatter coefficient and volume depolarization ratio and to identify the dust-related cirrus clouds (CALIPSO, 2023). The level-2 aerosol profile product (version 4.2) was used to obtain the vertical profiles of the aerosol extinction coefficient, particle depolarization ratio, and atmospheric volume description (Omar et al., 2009; CALIPSO, 2023). We can obtain the vertical feature mask (VFM), aerosol subtype, and cloud subtype to identify 'cirrus' and 'dust' (including dust and polluted dust) from the 'atmospheric volume description'. In addition, the meteorological data including the temperature, pressure, and relative humidity profiles along the satellite tracks were derived from the MEERA-2 reanalysis data (Gelaro et al., 2017) provided by the Global Modeling and Assimilation Office, which are integrated into the level-2 aerosol profile product.

### 2.2 ICNC derived from the DARDAR dataset

The DARDAR dataset is generated by a synergistic radar-lidar retrieval, combining the co-located and quasi-simultaneous detection of the CALIOP onboard CALIPSO and CPR onboard CloudSat, both of which belong to the 'A-train' constellation (Delanoë and Hogan, 2008, 2010). The DARDAR-Cloud product was utilized to give the parameters reflecting the properties of ice clouds, including the cloud extinction coefficient, cloud particle effective radius ($r_e$), and ice water content (IWC). The DARDAR-Nice profile product was employed to provide the profiles of in-cloud ICNC ($n_{ice}$) for ice crystals having diameters larger than 5, 25, and 100 µm, respectively (Sourdeval et al., 2018; Gryspeerdt et al., 2018; DARDAR, 2023). Both products possess a 60-m vertical resolution and a 1.7-km horizontal resolution. When lidar and radar observations are both available, the uncertainty in $n_{ice}$ is approximately 25%. Marinou et al. (2019) mentioned that high rates of homogeneous nucleation may cause a 50% underestimation of $n_{ice}$, attributed to deviations in the particle size distribution presumed.

### 2.3 Dust-related INPC obtained with POLIPHON method

The POLIPHON method was adopted to compute the dust-related INPC and dust mass concentration values (Tesche et al., 2009; Mamouri and Ansmann, 2015, 2016, 2017; Jing et al., 2023). It extracted the dust extinction coefficient $\alpha_d$ from the aerosol extinction coefficient $\alpha_p$ provided by CALIOP level-2 data using the one-step approach (Mamouri and Ansmann, 2014). Table 1 details the calculation process. The employed meteorological parameters are integrated within the CALIOP level-2 aerosol profile product. Here lidar ratio for dust $LR_d$ was assumed to be 45 sr (Peng et al., 2021; Floutsi et al., 2023). Then, the dust extinction coefficient can be converted into cloud-relevant parameters, including the dust mass concentration $M_d$, particle number concentration with a radius >250 nm $n_{250,d}$, and particle surface area concentration $S_d$ and $S_{100,d}$ (with a radius >100 nm), by multiplying the conversion factors $c_{v,d}$, $c_{250,d}$, $c_{s,d}$ and $c_{s,100,d}$. He et al. (2023) calculated the conversion factors over Midway Island (see figure 2). Ice nucleation in cirrus clouds involves both immersion and deposition freezing modes (Kanji et al., 2017; Marcolli, 2014). Here we disregarded contact freezing, which needs an INP to collide with a supercooled droplet. Therefore, for dust-related INPC computation, we applied the parameterization schemes of D10 (DeMott et al., 2010), D15 (DeMott et al., 2015), and U17-I (Ullrich et al., 2017) for immersion freezing as well as U17-D (Ullrich et al., 2017) for deposition freezing.



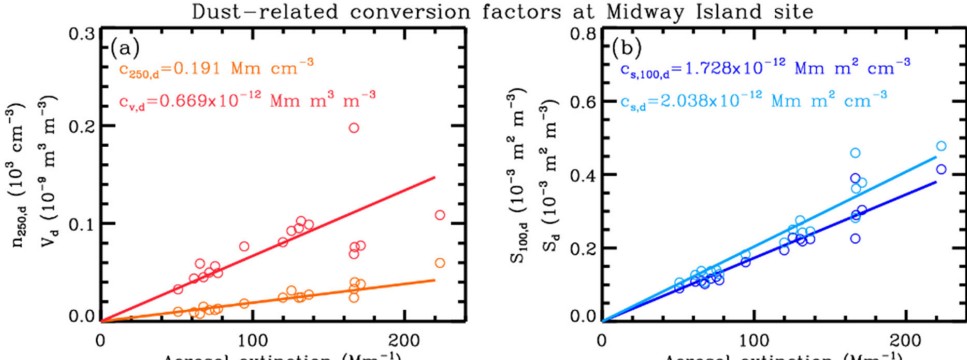

**Figure 2.** Relationship between the aerosol extinction coefficient and the volume concentration ($V_d$), particle number concentration with a radius >250 nm ($n_{250,d}$), particle surface area concentration $S_d$, and particle surface area concentration with a radius > 100 nm ($S_{100,d}$) over Midway Island. Each hollow circle denotes a data point, selected from the AERONET Version 3 database using the dust ratio threshold derived with the method given by Shin et al. (2019). The corresponding values of dust-related conversion factors $c_{v,d}$, $c_{250,d}$, $c_{s,d}$ and $c_{s,100,d}$ are given, respectively. For more detailed information, refer to He et al. (2023).

**Table 1.** Calculations and uncertainties for dust-related optical and ice-nucleating parameters (Tesche et al., 2009; Ansmann et al., 2019b). D10, D15, and U17 refer to the respective INP parameterizations in DeMott et al. (2010), DeMott et al. (2015), and Ullrich et al. (2017).

| Dust-related parameters | Computation | Uncertainty |
|---|---|---|
| Dust backscatter $\beta_d$ (Mm$^{-1}$ sr$^{-1}$) | $\beta_d(z) = \beta_p(z)\dfrac{(\delta_p(z) - \delta_{nd})(1 + \delta_d)}{(\delta_d - \delta_{nd})\left(1 + \delta_p(z)\right)}$ $\delta_d = 0.31, \delta_{nd} = 0.05$ | 10%–30% |
| Dust extinction $\alpha_d$ (Mm$^{-1}$) | $\alpha_d(z) = LR_d \times \beta_d(z)$ | 15%–25% |
| Dust mass conc. $M_d$ (µg m$^{-3}$) | $M_d(z) = \rho_d \times \alpha_d(z) \times c_{v,d}$ $\rho_d = 2.6$ g cm$^{-3}$ | 20%–30% |
| Particle number conc. (r>250 nm) $n_{250,d}$(cm$^{-3}$) | $n_{250,d}(z) = \alpha_d(z) \times c_{250,d}$ | 25%–35% |
| Particle surface area conc. $S_d$ (m$^2$cm$^{-3}$) | $S_d(z) = \alpha_d(z) \times c_{s,d}$ | 30 %–40% |
| D10 INP conc. $n_{INP}$ (L$^{-1}$) | $n_{INP}(p_0, T_0, T(z)) = a \cdot (273.16 - T(z))^b \cdot n_{250,d}(p_0, T_0)^{[c(273.16-T(z))+d]}$ $n_{INP}(z) = n_{INP}(p_0, T_0, T(z)) \cdot (T_0p(z))(p_0T(z))$ $a = 0.0000594; b = 3.33; c = 0.0265; d = 0.0033$ | 50 %–500% |
| D15 INP conc. $n_{INP}$ (L$^{-1}$) | $n_{INP}(p_0, T_0, T(z)) = f_d \cdot n_{250,d}(p_0, T_0)^{[a_d(273.16-T(z))+b_d]}$ $\cdot \exp[c_d(273.16 - T(z)) + d_d]$ $n_{INP}(z0 = n_{INP}(p_0, T_0, T(z)) \cdot (T_0p(z))(p_0T(z))$ $a_d = 0; b_d = 1.25; c_d = 0.46; d_d = -11.6; f_d = 3.0$ | 50 %–500% |
| U17-I(d) INP conc. $n_{INP}$ (L$^{-1}$) | $n_{INP}(z) = S_d(z) \times n_s(T(z))$ $n_s(T(z)) = \exp[150.577 - 0.517 \cdot T(z)]$ | 50 %–500% |
| U17-D(d) INP conc. $n_{INP}$ (L$^{-1}$) | $n_{INP}(z) = S_d(z) \times n_s(T(z), S_i)$ $n_s(T(z), S_i) = \exp\left\{a_u(S_i - 1)^{\frac{1}{4}}cos^2[b_u(T(z) - c_u)] \cot^{-1}[d_u(T(z) - e_u)/\pi]\right\}$ $a_u = 285.692; b_u = 0.017; c_u = 256.692; d_u = 0.080; e_u = 200.745$ | 50 %–500% |



### 2.4 MERRA-2 reanalysis data

The MERRA-2 reanalysis data are the latest version of global atmospheric reanalysis for the satellite era produced by NASA Global Modeling and Assimilation Office (GMAO) using the Goddard Earth Observing System Model (GEOS) version 5.12.4 (Gelaro et al., 2017). In this study, MERRA-2 data were utilized to provide the dust column density (GMAO, 2015). Two datasets covering the period from 1980 onwards were employed: (1) tavg1_2d_aer_Nx provides the daily total dust column mass density; (2) tavgM_2d_aer_Nx provides the monthly-mean dust column mass density.

### 2.5 HYSPLIT model

The HYSPIT (Hybrid Single-Particle Lagrangian Integrated Trajectory) model is a complete system widely used for atmospheric transport and dispersion, developed by the Air Resources Laboratory of NOAA (National Oceanic and Atmospheric Administration) (Stein et al., 2015). For a set starting time, an initial altitude, and geographical location, HYSPIT can compute either forward or backward trajectories of the air masses. The meteorological field from the Global Data Assimilation System (GDAS) archive is adopted in the calculation. In this study, the backward trajectories starting from Midway Island (28.21°N, 177.38°W) were stimulated to examine the potential regions of dust origin.

### 3. Results

When a cirrus cloud is embedded in or in contact with a dust layer, they spatially overlap with each other. In such cases, the dust layer and cirrus cloud are considered coupled, i.e., dust-cirrus interaction event (Ansmann et al., 2019a; He et al., 2021b). For heterogeneous nucleation, one INP primarily generates one ice crystal. For cirrus altitudes, Field et al. (2017) stated that secondary ice production scarcely occurred (takes place with temperatures ≥ -10°C). Therefore, a good agreement between INPC and ICNC would be expected if only heterogeneous nucleation is involved. Considering the uncertainties in retrieved INPC and ICNC, such a good agreement is considered within an order of magnitude (Ansmann et al., 2019a; Knopf et al., 2021). To explore whether transoceanic dust plumes can influence the cirrus formation over remote ocean regions via heterogeneous nucleation, here two cases occurring near Midway Island located in the central Pacific far from continents are studied in detail with the spaceborne observations by CALIOP and CPR. We emphasize examining the degree of agreement between dust INPC and in-cloud ICNC so that the dominant primary ice-nucleating mechanism within cirrus clouds can be well-determined.

### 3.1 Case on 5 May 2010

Figures 3a and b show the 532-nm total attenuated backscatter coefficient (TAB) and volume depolarization ratio $\delta_v$ on 5 May 2010 provided by the CALIOP level-1B product. An ice cloud appeared at altitudes of 8.3-10.0 km, which was indicated by $\delta_v$ exceeding 0.3 and strong TAB.  It was embedded in a dust layer, which had $\delta_v$ values of 0.1-0.2 and somewhat weaker TAB (in yellow). Figures 3c, d, and e show the vertical feature mask, cloud subtype, and aerosol subtype from the CALIOP level-2 aerosol profile product; cirrus cloud at altitudes of 8.3-10.0 km contacted with a dust layer at similar altitudes. Therefore, ice formation in the cirrus cloud was possibly influenced by dust particles via heterogeneous nucleation. Figure 4 shows the MERRA-2 dust mass column density on 5 May 2010 together with the 3-d backward trajectories computed by the HYSPLIT model. The dust plume originated from the Asian dust source regions two days earlier. It was constrained within an altitude range of 9.0-10.5 km and underwent advective transport to arrive in Midway Island on 5 May, which is consistent with the observations of CALIOP.




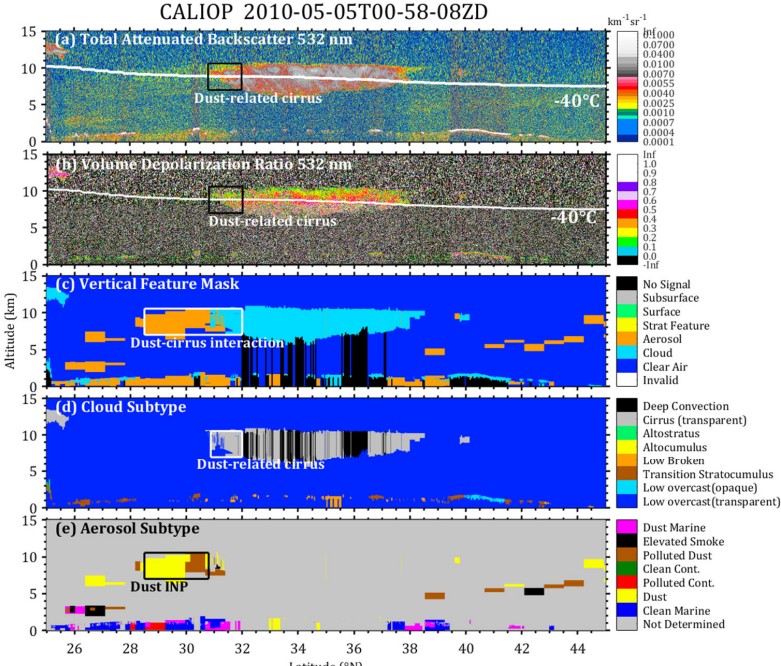

**Figure 3. CALIPSO altitude-orbit cross section of the CALIOP level-1B 532-nm (a) total attenuated backscatter coefficient, (b) volume depolarization ratio product, and CALIOP level-2 (c) vertical feature mask, (d) cloud subtype, and (e) aerosol subtype product on 5 May 2010. The corresponding orbit is 2010-05-05T00-58-08ZD.**

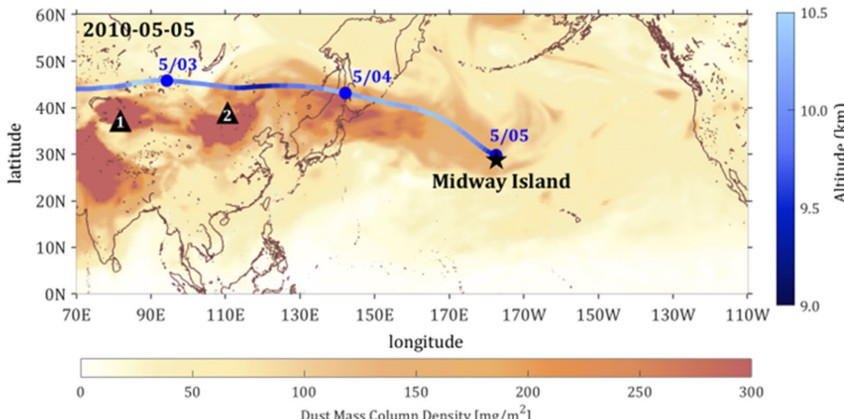

**Figure 4. The dust mass column density on 5 May 2010 from MERRA-2 data. The blue line denotes the 3-d backward trajectories starting from 3 May 2010 at altitudes of 9.8 km as simulated by the HYSPLIT model.**

Figure 5 shows the ice cloud properties, including cloud extinction coefficient, cloud particle effective radius, ice water content,
and ice crystal (with size <5 μm) number concentration ($n_{\text{ice,5 μm}}$) on 5 May 2010 from the DARDAR-Cloud product. The potential
dust-cirrus interaction region at 8.3-10.0 km is marked with a black rectangle (in figure 5b). Taking the cirrus cloud at latitudes of
30.8-32.0°N and altitudes of 8.3-10.0 km into consideration, the in-cloud average extinction coefficient, cloud particle effective
radius, and ice water content are 0.4 km$^{-1}$, 45.4 μm, and 10.0 mg m$^{-3}$, respectively. Figure 5d clearly shows that $n_{\text{ice,5 μm}}$ values




distribute unevenly within the latitude range marked by the black rectangle, which is thus divided into two parts for further analysis.

As respectively marked with black rectangles, part A appearing at latitudes of 31.5-31.6°N showed extremely high $n_{ice,5\ \mu m}$ values of >500 L$^{-1}$; while, part B appearing at latitudes of 31.6-31.8°N had much lower $n_{ice,5\ \mu m}$ values. Note that only the data points identified as cirrus with CALIOP VFM data and having valid data were used for calculation.

Figures 6a-d show the profiles of aerosol (dust and non-dust) extinction coefficient, particle depolarization ratio, large particles number (radius > 250 nm), surface area concentration, relative humidity, and temperature for the dust layers in the cloud-free area

(28.5-30.8°N). For the dust layer at altitudes of 9.0-9.8 km, the layer-average dust extinction coefficient is 25.6 Mm$^{-1}$, which reaches a similar level observed in previously reported cases in central China that is much closer to the Asian dust sources (He et al., 2022b), suggesting that a large number of dust particles can still be well-retained within the plume during their long-range transoceanic transport. The particle depolarization ratios generally range from 0.2 to 0.3, indicating that the dust plume is mainly composed of pure dust particles.

Figure 6e presents the dust-related INPC profiles for different freezing modes at corresponding temperature regions. At temperatures warmer than -35 ℃, the parameterization schemes D10, D15, and U17-I are adopted to calculate dust-related INPC; while at temperatures between -33 ℃ and -67 ℃, the parameterization U17-D is employed. At altitudes of 9.0-9.8 km, the layer-average dust-related INPCs for U17-D are 7.234 L$^{-1}$ (0.3-105.4 L$^{-1}$) for an assumed ice saturation ratio $S_i$ of 1.15, 96.3 L$^{-1}$ (3.4-1502.0 L$^{-1}$) for $S_i$ of 1.25, and 642.6 L$^{-1}$ (6.7-1.5×10$^4$ L$^{-1}$) for $S_i$ of 1.35. As a comparison, in-situ measurements over Florida

observed a slightly higher peak INPC of 300 L$^{-1}$ (at a temperature of 36 ℃ and RH$_i$ of 123%) in the dust layer transported from the Saharan desert (DeMott et al., 2003, 2009). In addition, INPC values reach a maximum at the top of the cirrus cloud and then gradually decrease as the altitude decreases since INPCs strongly depended on the temperature.

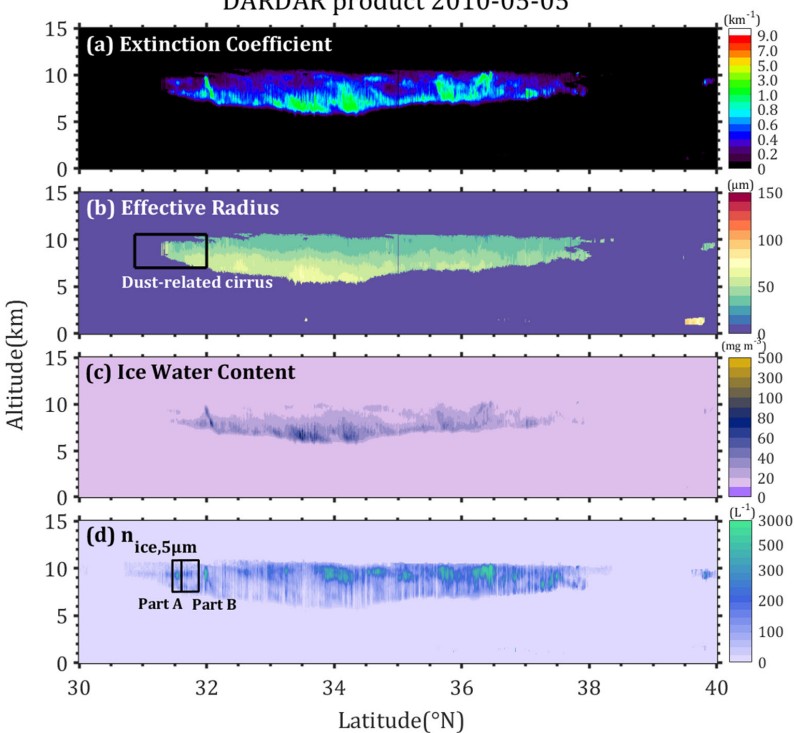

**Figure 5. Altitude-orbit cross section of the (a) cloud extinction coefficient, (b) cloud particle effective radius, (c) ice water content, and**

225 **(d) $n_{ice,5\ \mu m}$ from the DARDAR product on 5 May 2010. The corresponding orbit is 2010-05-05T00-58-08ZD.**

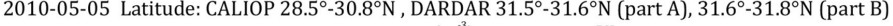

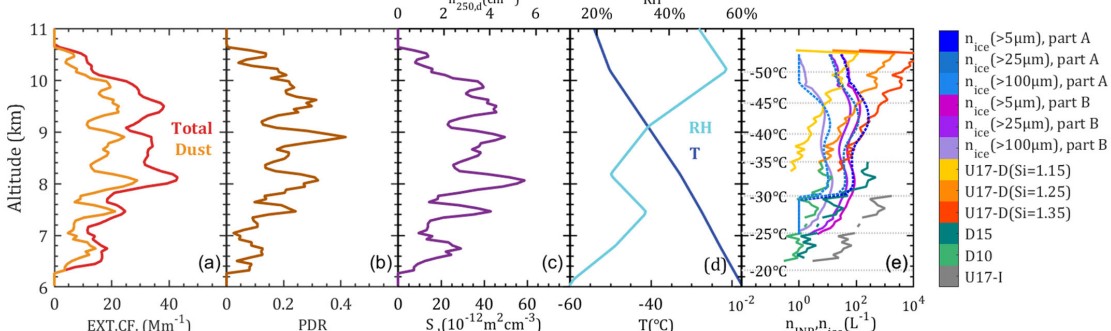

**Figure 6. Profiles of the 532-nm (a) dust and total (dust + non-dust) extinction coefficient, (b) particle depolarization ratio, (c) large particle number concentration (with radius >250 nm) $n_{250,d}$ and surface area concentration $S_d$, (d) relative humidity RH and temperature T (from MERRA-2 data), and (e) ice-nucleating particle concentration $n_{INP}$ (using parameterization schemes D10, D15, U17-D, and U17-I) and ice crystal number concentration $n_{ice}$ (from DARDAR-Nice product) on 5 May 2010. Profiles of INP-related parameters are calculated by merging the CALIOP measurements between 28.5-30.8°N). For the DARDAR-Nice product, the integrated profiles are within the latitude range of 31.5-31.6°N for part A and 31.6-31.8°N for part B. $S_i$ denotes the ice saturation ratio.**

Figure 6e also provides the ice crystal number concentrations larger than 5, 25, and 100 μm obtained from the DARDAR-Nice product, which are denoted as $n_{ice,5 \mu m}$, $n_{ice,25 \mu m}$ and $n_{ice,100 \mu m}$, respectively. All these ICNC values show an enhancement around an altitude of 9.5 km, which is approximately 0.5 km below the cloud top. In general, pristine ice crystals nucleate near the top of the cloud, which is the coldest part due to radiative cooling and has the most abundant INP supply; nucleated ice crystals begin to sediment once they grow up to sizes of 50-100 μm (Heymsfield et al., 2017; Ansmann et al., 2019a), resulting in the presence of ICNC enhancement slightly below the cloud top, which is especially distinct for large-size ICNC ($n_{ice,100 \mu m}$). For part A within the latitude of 31.5-31.6°N and the altitude of 9.0-9.6 km, the average ICNCs within the cirrus clouds are 209.3 L⁻¹ (66.1-485.5 L⁻¹) for $n_{ice,5 \mu m}$, 92.4 L⁻¹ (27.8-207.0 L⁻¹) for $n_{ice,25 \mu m}$, and 8.3 L⁻¹ (1.2-16.1 L⁻¹) for $n_{ice,100 \mu m}$. The ICNC values at this altitude range are rather high (>300 L⁻¹) with a maximum $n_{ice,5 \mu m}$ of 485.5 L⁻¹, revealing that both homogeneous may also take place in this region accompanied by dust-triggered heterogeneous nucleation.

For part B within 31.6-31.8°N, the average in-cloud ICNC values are 111.2 L⁻¹ (30.5-249.3 L⁻¹) for $n_{ice,5 \mu m}$, 51.7 L⁻¹ (14.2-111.8 L⁻¹) for $n_{ice,25 \mu m}$, and 6.6 L⁻¹ (1.7-11.5 L⁻¹) for $n_{ice,100 \mu m}$, respectively. Ansmann et al. (2019a) observed ICNC values of 4.3-39 L⁻¹ in cirrus clouds. Cziczo et al. (2013) reported that the ICNC values for heterogeneous freezing are typically 1-100 L⁻¹. Thus, the ICNC values observed in this case reflect the typical situation of heterogeneous nucleation within the cirrus clouds. At the upper troposphere above 9 km, the cold part of the cirrus clouds with temperatures below -40 °C, homogeneous nucleation is generally considered to take place because it is usually very clean there. However, in this case, abundant dust INPs were provided by the long-range transpacific dust plume passing by. Therefore, when dust particles came across this region with a high moisture level, it is conducive to the occurrence of heterogeneous nucleation and subsequent suppression of homogeneous nucleation by consuming available water vapor (Kärcher et al., 2022). According to the quantificational comparison between the ICNC and INPC values, it is found that U17-D-derived INPCs with $S_i$ of 1.15 well agree with in-cloud $n_{ice,5 \mu m}$ (ICNC-to-INPC ratio of 0.9). Additionally, U17-D-derived INPCs with $S_i$ of 1.25 are closer to $n_{ice,25 \mu m}$ (ICNC-to-INPC ratio of 0.5). Considering that the ICNC and INPC values are substantially consistent with each other within an order of magnitude, it can be inferred that heterogeneous freezing can solely be responsible for ice formation in the cirrus cloud.



In this case, it is a cirrostratus with a very large horizontal extent ranging from a latitude of 31°N to 38°N. The mechanisms of ice formation for part A and part B are diverse. Homogeneous nucleation also took place to compete with heterogeneous nucleation in part A, while it is probable that heterogeneous nucleation dominantly occurred in Part B. As seen from figure 5d, typical ICNC values for homogeneous nucleation (in green) are intermittently observed a few hundred meters below the cloud top, revealing that the supply of dust INPs was not uniform within the whole cloud layer. We conjecture that dust particles were unevenly entrained into a high moisture region before cloud formation. It is of significance to affirm that transoceanic dust particles may indeed promote cirrus clouds to form via heterogeneous nucleation even over the central Pacific.

**3.2 Case on 27 April 2008**

Another case was observed on 27 April 2008 as shown in figure 7. A series of ice clouds at altitudes of 10.0-11.4 km appeared between 27.1°N and 28.3°N, as marked by black rectangles, and showed relatively intense TAB and a large $\delta_v$ of > 0.3 (figures 7a and b). Figure 7b shows that the ice clouds were embedded in a dust plume located at the same altitudes with $\delta_v$ values of 0.1-0.2 and a relatively weaker TAB. Figures 7c, d, and e show the VFM, cloud subtype, and aerosol subtype. 'Cirrus clouds' can be seen with 'dust' and 'polluted dust' in the vicinity (both lateral sides). Moreover, an elevated smoke layer at lower altitudes (8-10 km) was also observed to the north of the abovementioned dust layer, which may have a potential impact on the ice formation within the wide range of the dense clouds at latitudes of 29-32°N. Figure 8 shows the MERRA-2 dust mass column density on 27 April 2008 together with the 5-d backward trajectories computed by the HYSPLIT model. The dust plume originated from the Asian dust source regions three days earlier and arrived over Midway Island on 27 April. The dust particles were mainly constrained at altitudes of 7.0-10.0 km during the long-range transport.

Figure 9 shows the ice cloud properties provided by the DARDAR-Cloud product, including the cloud extinction coefficient, cloud particle effective radius, ice water content, and ice crystal (with size <5 μm) number concentration ($n_{ice,5\,\mu m}$). The specific time and location are the same as the data presented by CALIOP products in figure 7. Considering the cirrus cloud at 10.0-11.4 km and latitudes of 27.1-28.3°N, the in-cloud average extinction coefficient, cloud particle effective radius, and ice water content are 0.6 km⁻¹, 43.8 μm, and 15.2 mg m⁻³, respectively. In figure 9d, two specific parts of clouds are selected according to their distinct ICNC values, as denoted by part A and part B. Part A extended from 27.8°N to 28.0°N with very large ICNC values, while part B at the range of 28.0-28.2°N showed much smaller ICNC values. Note that these average values were calculated solely based on the 'cirrus' data points identified (using the CALIOP cloud subtype).

The dust-presence CALIOP aerosol extinction coefficient and particle depolarization ratio profiles within latitudes of 27.1-28.3°N were utilized to estimate the effective concentration of dust INPs nearby the cirrus clouds as shown in figure 9. For the dust layer at 10.2-11.4 km, $\delta_p$ takes a relatively small peak value of 0.15 at an altitude of 10.5 km, indicating that most coarse dust particles with larger $\delta_p$ (Sakai et al., 2010) have probably removed via sedimentation before arriving over Midway Island; while the dust extinction coefficient is on average 10.6 Mm⁻¹ with a maximum of 33.7 Mm⁻¹ also at 10.5 km. In addition, comparing the total and dust extinction coefficient, it is noticed that there is also a mass of non-dust aerosols carried within the dust plume, which may be another reason explaining the low $\delta_p$ values in this case. Smoke can be a possible type of aerosol that was mixed into the dust layers (see figure 7e).



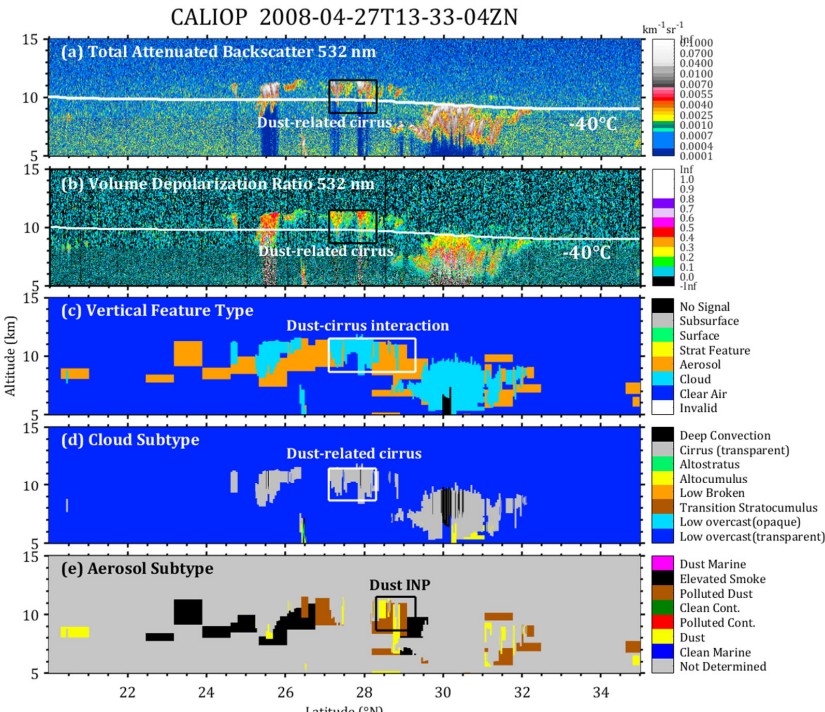

**Figure 7. CALIPSO altitude-orbit cross section of the CALIOP level-1B 532 nm (a) total attenuated backscatter coefficient, (b) volume depolarization ratio product, and CALIOP level-2 (c) vertical feature mask, (d) cloud subtype, and (e) aerosol subtype product on 27 April 2008. The corresponding orbit is 2008-04-27T13-33-04ZN.**

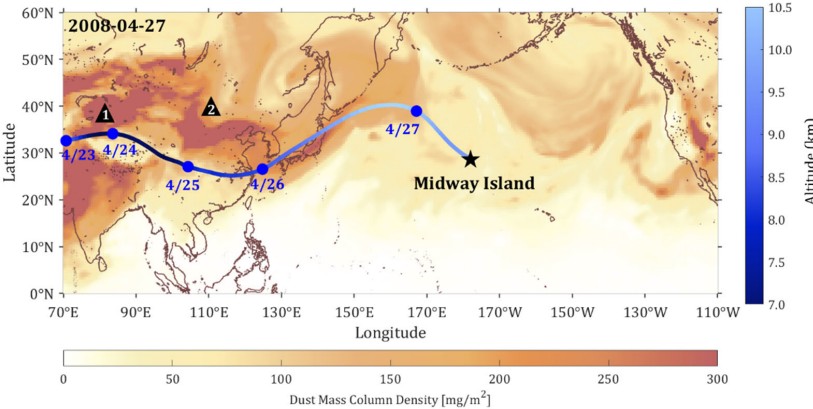

**Figure 8. The dust mass column density on 5 May 2010 from MERRA-2 data. The blue line denotes the 5-d backward trajectories starting from 23 April 2008 at altitudes of 8.5 km as simulated by the HYSPLIT model.**

The profiles of dust mass concentration, large particle (with radius >250 nm) number concentration, surface area concentration, dust-related INPC, and in-cloud ICNC are shown in figure 10. Cirrus clouds generally appeared at altitudes of 10.2-11.4 km with temperatures ranging from -33 °C to -67 °C. U17-D parameterization for deposition freezing was used to calculate dust-related INPCs at cirrus altitudes; the ice saturation ratio $S_i$ were assumed to be 1.15, 1.25, and 1.35. In addition, for immersion mode, the parameterization schemes D10, D15, and U17-I were also used for calculating the INPC below the -35°C isotherm. The average



dust-related INPCs for U17-D with $S_i$ of 1.15, 1.25, and 1.35 are 27.0 L$^{-1}$ (0.1-298.6 L$^{-1}$), 416.7 L$^{-1}$ (0.8-4800.0 L$^{-1}$), and 3101.0 L$^{-1}$ (5.3-3.7×10$^4$ L$^{-1}$), respectively.

The in-cloud average ICNC values show a distinct disparity between part A and part B. For part A, the average ICNC values are 634.6 L$^{-1}$ (4.6-3458.0 L$^{-1}$) for $n_{ice,5\,\mu m}$, 277.8 L$^{-1}$ (2.0-1296.0 L$^{-1}$) for $n_{ice,25\,\mu m}$, and 30.1 L$^{-1}$ (0.1-105.9 L$^{-1}$) for $n_{ice,100\,\mu m}$, respectively. Similar to figure 9d, an evident enhancement of ICNC values appeared at altitudes of 10.2-11.4 km and reached the peak at approximately 11.2 km as seen in figure 10e. Part A substantially showed $n_{ice,5\,\mu m}$ of larger than 500 L$^{-1}$. Such high ICNC values are typically associated with homogeneous nucleation, revealing the competition between heterogeneous and homogeneous nucleation in part A. In addition, comparing the dust-related INPCs for U17-D with $S_i$ of 1.35 with $n_{ice,5\,\mu m}$, the ICNC-to-INPC ratio farther exceeds one order of magnitude, meaning that homogeneous nucleation dominated the ice nucleation in part A.

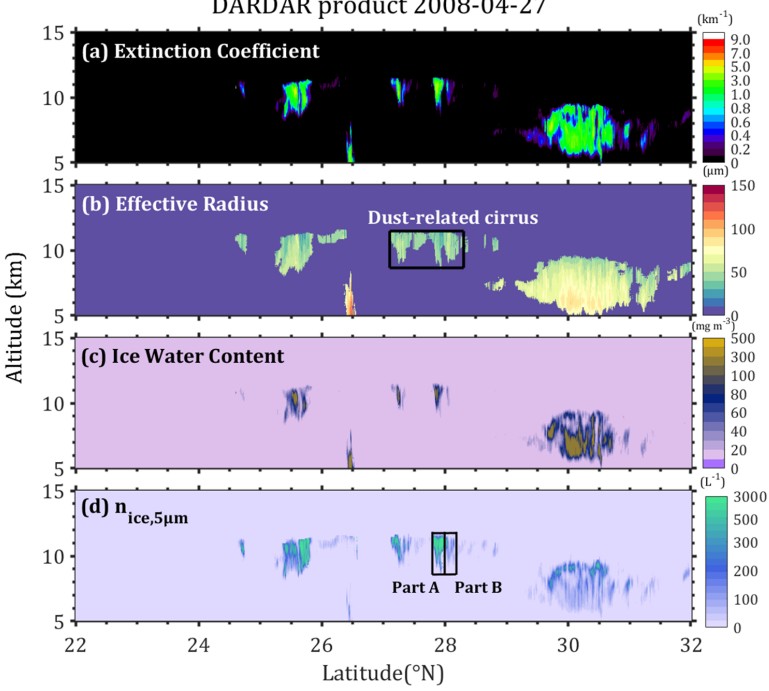

**Figure 9. Altitude-orbit cross section of the (a) cloud extinction coefficient, (b) cloud particle effective radius, (c) ice water content, and (d) $n_{ice,5\,\mu m}$ from the DARDAR product on 27 April 2008. The corresponding orbit is 2008-04-27T13-33-04ZN.**

For part B, the average ICNC values are 79.9 L$^{-1}$ (8.2-510.9 L$^{-1}$) for $n_{ice,5\,\mu m}$, 39.4 L$^{-1}$ (3.6-227.8 L$^{-1}$) for $n_{ice,25\,\mu m}$ and 7.6 L$^{-1}$ (0.1-47.5 L$^{-1}$) for $n_{ice,100\,\mu m}$. Part B mainly shows relatively lower ICNC values of <300 L$^{-1}$. By comparing the ICNC and INPC values, figure 10e shows that the U17-D-derived INPCs with $S_i$ of 1.15 agree well with the in-cloud $n_{ice,25\,\mu m}$ with an ICNC-to-INPC ratio of 1.5. As a result, we conclude that only heterogeneous nucleation may explain the ice formation in part B.

As seen from the smaller horizontal extent, the cirrus cloud is more likely a cirrocumulus that is different from the stratocirrus in the first case. Similar to the first case, it is also found that ice nucleation mechanisms in part A and part B are significantly different from each other. The $n_{ice,5\,\mu m}$ and $n_{ice,25\,\mu m}$ values in part A are much larger than those in part B, which can only be explained by the participation of homogeneous freezing that produced more but smaller ice crystals (Liu et al., 2012). For part B, much lower ICNC values and good agreement between in-cloud ICNC and nearby dust-related INPC both support the sole occurrence of heterogenous nucleation triggered by dust particles. Nevertheless, these occasionally-appeared lower ICNC values confirm the transoceanic dust particles are non-negligible in cirrus formation via heterogeneous nucleation even over the central Pacific.

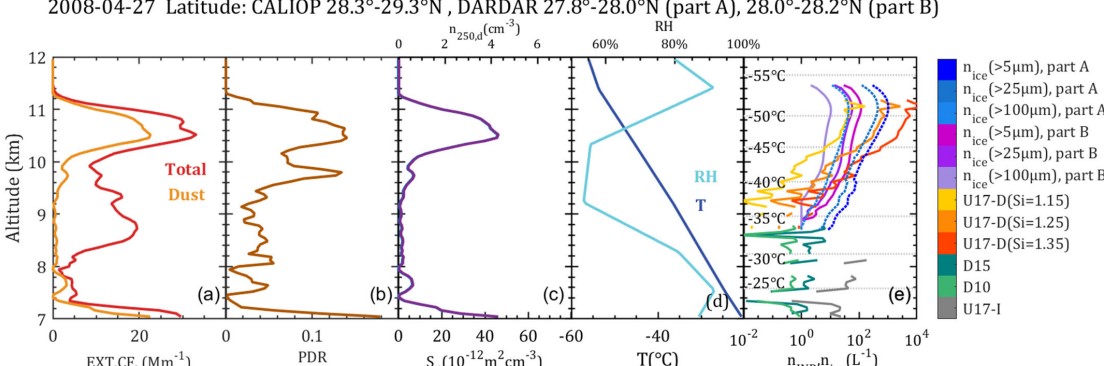

**Figure 10. Profiles of the 532-nm (a) dust and total (dust + non-dust) extinction coefficient, (b) particle depolarization ratio, (c) large particle number concentration (with radius >250 nm) $n_{250,d}$ and surface area concentration $S_d$, (d) relative humidity RH and temperature T (from MERRA-2 data), and (e) ice-nucleating particle concentration $n_{INP}$ (using parameterization schemes D10, D15, U17-D, and U17-I) and ice crystal number concentration $n_{ice}$ (from DARDAR-Nice product) on 27 April 2008. Profiles of INP-related parameters are calculated by merging the CALIOP measurements between 28.3-29.3°N). For the DARDAR-Nice product, the integrated profiles are within the latitude range of 27.8-28.0°N for part A and 28.0-28.2°N for part B. $S_i$ denotes the ice saturation ratio.**

## 4. Discussions and Conclusions

Cirrus clouds play a vital role in global climate by regulating the radiative balance of the Earth, which is essentially determined by the microphysical properties of ice crystals within cirrus and is further associated with the ice-nucleating regimes, i.e., homogeneous nucleation or heterogeneous nucleation. To examine the potential influence of long-range transport dust on the cirrus formation over remote oceans, two cases regarding dust-cirrus interactions are studied over the central Pacific near Midway Island (28.21°N, 177.38°W) in detail. A previously proposed method (He. et al., 2022b) is adopted to examine the ice-nucleating mechanism of two cirrus cloud cases based on spaceborne observations. To perform the identification, i.e., comparing in-cloud ICNC with the adjacent dust-related INPC, we follow a series of steps to obtain the requisite data. The DARDAR dataset was utilized to obtain the in-cloud ICNC. Then, the CALIOP observational data together with the POLIPHON method was employed to derive the dust-related INPC. In the POLIPHON method, the dust-related conversion factors for Midway Island are provided by He et al. (2023).

The overview of the observational results of the two cases is presented in Table 2. The first case on 5 May 2010 showed a stratocirrus with a very large horizontal extent ranging from a latitude of 31°N to 38°N. The ICNC values in part A reached a fairly high of >300 L⁻¹ ($n_{ice,5\mu m}$), meaning that homogeneous nucleation may participate in producing ice crystals. For part B, the dust-related INPC and ICNC values agree well with each other within an order of magnitude, indicating the sole occurrence of heterogeneous nucleation. Typical ICNC values for homogeneous nucleation intermittently observed in the cloud layer reflect that the supply of dust INPs was not uniform and dust particles were unevenly entrained into the cloud formation regions with high moisture. In the cloud parcel without sufficient INP supply, homogeneous nucleation can be dominated. The second case on 27 April 2008 is more likely a cirrocumulus with a much smaller horizontal extent. The ICNC values as high as >500 L⁻¹ ($n_{ice,5\mu m}$) are observed in part A, suggesting that homogeneous nucleation also took part in ice formation so that a surging number of small ice crystals were produced. Part B in the second case also shows the sole occurrence of heterogeneous nucleation as seen from the coincident of ICNC and dust-related INPC values. Both cases affirm that transoceanic dust particles may indeed promote cirrus clouds to form via heterogeneous nucleation even over the central Pacific, which is of significance for cirrus formation in remote ocean areas.





**Table 2. Overview of in-cloud ICNC and dust INPC for the two cases on 5 May 2010 and 27 April 2008 near Midway Island. The ICNC values are provided by the DARDAR Nice product. The INPC values are retrieved from the CALIOP dust extinction coefficient based on the POLIPHON method. The layer-average values for ICNC and INPC are given together with the minimum and maximum values in parentheses. $S_i$ denotes the ice saturation ratio.**

| Parameter | 5 May 2010 | | 27 April 2008 | |
|---|---|---|---|---|
| | Part A | Part B | Part A | Part B |
| Ice-nucleating type | homogeneous and heterogeneous nucleation | Solely heterogeneous nucleation | homogeneous and heterogeneous nucleation | Solely heterogeneous nucleation |
| Latitude of cirrus (°N) | 31.5-31.6 | 31.6-31.8 | 27.8-28.0 | 28.0-28.2 |
| ICNC, $n_{ice,5\mu m}$ (L$^{-1}$) | 209.3 (66.1-485.5) | 111.2 (30.5-249.3) | 634.6 (4.6-3458.0) | 79.9 (8.2-510.9) |
| ICNC, $n_{ice,25\mu m}$ (L$^{-1}$) | 92.4 (27.8-207.0) | 51.7 (14.2-111.8) | 277.8 (2.0-1296.0) | 39.4 (3.6-227.8) |
| ICNC, $n_{ice,100\mu m}$ (L$^{-1}$) | 8.3 (1.2-16.1) | 6.6 (1.7-11.5) | 30.1 (0.1-105.9) | 7.6 (0.1-47.5) |
| Altitude of cirrus (km) | 9.0-9.8 | | 10.2-11.4 | |
| Temperature of cloud (°C) | -40.1 to -46.8 | | -44.4 to -53.8 | |
| Altitude of dust layer (km) | 9.0-9.8 | | 10.2-11.4 | |
| Latitude of dust layer (°N) | 28.5-30.8 | | 28.3-29.3 | |
| Dust INP, U17-d, $S_i$=1.15 (L$^{-1}$) | 7.2 (0.3-105.4) | | 27.0 (0.1-298.6) | |
| Dust INP, U17-d, $S_i$=1.25 (L$^{-1}$) | 96.3 (3.4-1502.0) | | 416.7 (0.8-4765.0) | |
| Dust INP, U17-d, $S_i$=1.35 (L$^{-1}$) | 642.6 (6.7-1.5×10$^4$) | | 3101.0 (5.3-3.7×10$^4$) | |

He et al. (2022b) observed that transported Asian dust can act as INPs to influence cirrus formation over central China and clarified the ice nucleation regime by comparing the in-cloud ICNC and dust INPC. Based on the same method, this study performs two case studies and prove that long-range transoceanic Asian dust plume can influence cirrus formation much farther over the central Pacific by providing a much higher level of INPC (generally an order of magnitude higher) than those in He et al. (2022b). The huge reservoir of INPs in the two cases originated from more intense Asian dust events, in which the Mongolian anticyclones lifted up a much larger amount of dust particles to initiate long-range transport. The study here reveals that the ice formation and microphysical properties of cirrus clouds over remote ocean regions can be regulated by the natural sources of INPs. At the upper troposphere, besides the long-range transported dust aerosols, sea spray particles originating from the sea surface (Twohy and Poellot, 2005; Patnaude et al., 2021), smoke aerosols (Raga et al., 2022; Mamouri et al., 2023) emitted during wildfire events, and volcanic aerosols (Friberg et al., 2015; Sporre et al., 2022) from occasional but intense eruptions may also have a potential impact on the ice-nucleating regime and radiative forcing of cirrus clouds over remote oceanic regions. Without these natural INP supply, cirrus clouds would not form since relative humidity with respect to ice rarely reaches up to 140-150% over clear-sky conditions (Cziczo et al., 2013; Dekoutsidis et al., 2023), which is however necessary for homogeneous nucleation. In other words, the natural supply of INPs to the upper troposphere may increase the cloud cover to reflect more solar radiation over oceanic regions and modulate the microphysical properties of cirrus clouds by differentiating ice-nucleating regimes, both of which may cause a cooling effect on global climate. It is suggested that these aerosol-cirrus interactions over remote oceanic regions should be well considered in climate evaluation and transoceanic dust aerosols must be quantificationally estimated from an angle of 3-D view.

A number of following-up works should be done in the future. This study only selects two clear-cut cases, and thus, a statistical study with long-term observational data is needed to further examine the influence of dust-cirrus interactions on global climate. Also, other transoceanic pathways of dust plumes should be studied in detail, such as the transatlantic dust from North Africa (Yu et al., 2021), the Arctic dust transported from high-latitude dust sources of the North Hemisphere (Bullard et al., 2016; Meinander et al., 2022) and Asia (Huang et al., 2015), the Northern Atlantic dust from Iceland (Baddock et al., 2017), and the long-range transported dust over the ocean of the Southern Hemisphere from Australia, Patagonia, and Southern Africa (Struve et al., 2020; Kok et al., 2021; Meinander et al., 2022). This may help us to better evaluate the impact of aerosol-cloud interactions, which is still the most important contributor to the uncertainties in global radiative forcing (IPCC, 2021).



**Data availability**

CALIPSO data used in this work can be accessed through the website https://subset.larc.nasa.gov (CALIOP, 2023). DARDAR products are at the website https://www.icare.univ-lille.fr (DARDAR, 2023). MERRA-2 reanalysis data can be obtained via https://doi.org/10.5067/LTVB4GPCOTK2 (GMAO, 2015). HYSPLIT model are available at the website https://www.arl.noaa.gov. AERONET data over Midway Island can be obtained at https://aeronet.gsfc.nasa.gov/new_web/data.html (AERONET, 2023).

**Author contributions**

HS analyzed the data and wrote the manuscript. ZY participated in scientific discussions and reviewed and proofread the manuscript. YH conceived the research, acquired the research funding, and wrote and reviewed the manuscript. YZ and LW reviewed the manuscript and participated in scientific discussions. DJ analyzed the data.

**Competing interests**

The contact author has declared that none of the authors has any competing interests.

**Financial support**

This work has been supported by the National Natural Science Foundation of China (grant nos. 42005101, 42205130, and 62105248), the Chinese Scholarship Council (CSC) (grant no. 202206275006), the Meridian Space Weather Monitoring Project (China), and the Innovation and Development Project of China Meteorological Administration (grant no. CXFZ2022J060).

**Acknowledgements**

The authors thank the Atmospheric Science Data Central (ASDC) at the NASA Langley Research Center for providing the CALIPSO data (https://subset.larc.nasa.gov/calipso/login.php), the ICARE Thematic Center for generating and storing the DARDAR products (https://www.icare.univ-lille.fr), the Global Modeling and Assimilation Office (GMAO) for the MERRA-2 reanalysis data (https://fluid.nccs.nasa.gov/reanalysis/chem2d_merra2) (GMAO, 2015), and the NOAA Air Resources Laboratory
(ARL) for the HYSPLIT model (https://ready.arl.noaa.gov/HYSPLIT_traj.php).

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
