# Peer review of "Measurement report: Influence of long-range transported dust on cirrus cloud formation over remote ocean: Case studies near Midway Island, Pacific"

_EGUsphere, 2023_

## Referee Comment (RC4)

**Review of 'Measurement report: Influence of long-range transported dust on cirrus cloud formation over remote ocean: Case studies near Midway Island, Pacific'**

**Principal criteria:**

Scientific significance: Fair (2)
Scientific quality: Good (3)
Presentation quality: Fair (2)

The manuscript by Shen *et al.* presents an observational study of cirrus cloud formation near Midway Island in Pacific Ocean and the ice-nucleation properties of long-range transported desert dust. Authors have used observational data from Cloud-Aerosol Lidar with Orthogonal Polarization (CALIOP) and Cloud Profiling Radar (CPR), DARDAR and MERRA-2 datasets, and POLIPHON and HYSPLIT models for the data evaluation. As is, the manuscript addresses an important topic (especially for atmospheric and climate modeling communities) – the impact of long-range transport dust on the formation of cirrus clouds – but does not present an answer. Before the paper is published, some major concerns would have to be addressed.

My greatest concern with the manuscript is the lack of indepth analysis – additional details (such as the nucleation mechanism, connection between the initial nucleation conditions and lab experiments) would improve the manuscript significantly. For this reason, these Major issues in the manuscript should be addressed:

1. The idea and dataset chosen to show the nucleation and cloud formation are great, but the data analysis seems to be focused on an observation. It would be very interesting to see the atmospheric conditions data and the discussion of the initial heterogeneous ice nucleation (e.g., the transition from pure dust to pure cloud at 30.34/0.0 lat/lon on 2010-05-05 trajectory) to see the relation to the laboratory experiments (such as Koehler et al., Atmos. Chem. Phys., 10, 11955–11968, 2010).

2. The manuscript is presented as a Measurement Report, however it is not a new or original measurement, but rather a reanalysis of an old and public dataset. As a result, the manuscript does not provide substantial insight and conclusions. I would suggest the authors to perform and write-up an in-depth analysis of the processes based on their expertise and reconsider the manuscript as a research article.

3. Another major shortcoming of the article is the Results section, where most of the section is focused on spelling out the results seen in figures with little interpretation. It is good to see that the summary of the results has been provided in Table 2, in the Discussion section, but I would expect it as part of Results.

For these reasons, in my view, this work is not yet sufficient for publication and I would reconsider the manuscript after major revisions.

Minor issues:

4. In Abstract, lines 24 and 25, the text says '[...] nucleation is dominated [...]' and '[...] nucleation can still be dominated [...]' while it should be 'dominant'.

5. Line 70, 'concerned' should be 'considered'?

6. Mistypes and inconsistent labeling of the instrument and datasets: sections 2.1: MEERA-2; 2.5: HYSPIT.

7. Missing satellite track, it should be added to the lon/lat maps. Vertical profile figures should have double latitude + longitude axis on abscissa (same as provided by CALIPSO).

8. The results section is riddled with 'Figure <n> shows [...]' sentences. I would suggest rephrasing them to 'As seen in Fig. <n>[...]', 'Based on data shown in Fig. <n>[...]', etc.

9. Throughout the manuscript there are numerous grammatical errors (similar to the ones pointed out above) and it should be very carefully revised.

10. The authors have a significant number of self-citations that are present in addition to other previous works, e.g. (He et al., 2021b, 2022b), (Jing et al. 2023). I would suggest removing them where appropriate.

11. The references (He et al., 2021a) and (He et al., 2022c) are not mentioned in the text at all. Please make sure that all unused references are removed.

12. The abstract provides a good summary, but is quite verbose and lengthy. If it is possible to shorten it without affecting quality, it would be excellent.

13. In Figure 8, the dust mass column density is for 2010-05-05, but the HYSPLIT trajectories span from 2008-04-23 to 2008-04-28. Is it a mistake in the caption, or was wrong dataset used for the figure?

---

## Author Comment (AC1)

**Response to Reviewer #1**

**General Comments:**

This is an interesting study that relates mineral dust plumes from Asian deserts to the number concentration of ice particles in cirrus clouds over Midway Island in the Central Pacific. It does this strictly through satellite remote sensing, using retrieval methods for mineral dust concentration (i.e., ice nucleating particle concentration or INPC) and ice particle number concentration (ICNC). It is well written and organized, but some of the arguments do not appear to be supported by the data, and the results may be overinterpreted. Specifics are given below. I recommend major revisions in rewriting the article, but this may not require a great deal of additional work.

**Response:** We are grateful for the reviewer's constructive comments on the manuscript. All the comments have been addressed in the revised manuscript, and the responses to each comment are given below. Therefore, the specific discussions have been largely updated now. The main revisions are listed below:

(1) As strongly suggested by reviewer #2, here we decide to combine Part A and Part B and conduct the analysis by considering them as a whole, due to the uncertainty of DARDAR retrievals. The specific discussions have been largely updated now.

(2) As at least three reviewers suggest shifting this manuscript from 'Measurement Report' to 'Research Article', we would like to do so but also need to involve Handling Editor Prof. Krämer in the decision. In addition, considering the results have been largely rewritten by adding in-depth discussions, we also think it would be better to change it to 'Research Article'.

(3) As the cirrus clouds in the two cases have already formed for at least half an hour (as deduced from the vertical extent of the ice virga by assuming a falling velocity of 1 m/s), nucleated ice crystals may have undergone significant growth. Therefore, in the comparison between ICNC and INPC, we decide to mainly use the values of $n_{ice,25um}$ and $n_{ice,100um}$. The specific reason has been discussed in the text of the revised manuscript.

(4) As there is no evident indication of the depletion of dust INP, the possibility for occurring homogeneous nucleation is low. Therefore, we have removed all the discussions about the involvement of homogeneous nucleation in the cirrus clouds.

**Specific Comments:**

**Comments 1:** Figure 4: The high column dust density measurements shown here in the back-trajectory do not ensure that air overlying these Asian deserts at ~ 10 km is having a relatively high concentration of mineral dust, since the column dust magnitude could result almost entirely from dust much below 10 km. This needs to be stated. On the other hand, it is commendable that the authors did this analysis since, although incomplete, it may be using all the available data and does provide important information.

**Response 1:** According to the suggestion, we have added the statements to mention that the MERRA-2 column dust density may be mainly contributed by the relatively high concentration of mineral dust at relatively lower levels. (Please see lines 197-198) Nevertheless, we just would like to employ the MERRA-2 column dust density to examine the rough transport pathway and possible original region of the dust plumes from a large-scale view.

**Comments 2:** Figure 5: There appears to be a problem with color legend. The plotting background is violet, which corresponds to an ice water content of 10-20 mg/m3. Not possible. The same problem occurs with Fig. 9c.

**Response 2:** Thank you for pointing out this. We have updated the color bars for Figures 5c and 9c.

**Comments 3:** Lines 234-235: For what purpose are ICNCs shown for $n_{ice,25um}$ and $n_{ice,100um}$? They provide

no closure information with respect to dust INP concentration since most of the ICNC can be associated with D < 25 um (Kramer et al., 2009, ACP). Only $n_{ice,5um}$ is relevant to the closure being sought with INPC.

**Response 3:** Thanks for the reviewer's comment. As suggested by reviewer #2, it is better to retain all of the information on $n_{ice,5um}$, $n_{ice,25um}$, and $n_{ice,100um}$, not only for the closure with respect to dust INPC but also for '*getting a better feeling for the large uncertainties in all ICNC products* (as commented by reviewer #2)'.

**Comments 4:** Lines 236-237: Suggest citing Diao et al. (2015, JGR) to back up this statement.

**Response 4:** We have added it to the revised manuscript.

**Comments 5:** Lines 252-253: Figure 6e shows agreement between ICNC and INPC only near cloud top for $S_i = 1.15$, where ICNC is for $n_{ice,5um}$. From the previous page, it states that the layer average INPC is 7 $L^{-1}$ and 96 $L^{-1}$ for $S_i$ of 1.15 and 1.25, respectively. Since the layer average ICNC is 111 $L^{-1}$ for Part B, optimal agreement is for $S_i = 1.25$, with ICNC-to-INPC ratio of 1.16. Therefore, $S_i$ here should be 1.25, not 1.15.

**Response 5:** We have modified the way we delineate our study regions by combining both Part A and Part B as a whole. As a result, the CNC-to-INPC ratios have been updated. Now the results part has been largely modified.

**Comments 6:** Line 253: It is not clear how the ICNC-to-INPC ratio can be 0.9 for $S_i = 1.15$, based on the previous text. As noted above, this ratio appears to be 1.16 for $S_i = 1.25$, for which closure is optimal.

**Response 6:** As explained in Response 5, we have made the necessary modifications after combining Part A and Part B as a whole in the revised manuscript. Thus, we would like the reviewer can reevaluate the analysis.

**Comments 7:** Lines 317-320: Since a large portion of the ICNC resides at D < 25 μm, $n_{ice,5um}$ should be used rather than $n_{ice,25um}$ (as assumed in this study). This practice (of using $n_{ice,5um}$) was followed for the other May 5th case study. In Fig. 10e, $S_i = 1.25$ agrees best with $n_{ice,5um}$, consistent with the previous case study based on Comment 5 above.

**Response 7:** As previously mentioned in Response 5, we have revised the specified study areas. Also in both cases, the ice virgae exceed 2 km, which indicates that they have to be formed at least half an hour before (reasonably assuming a falling velocity of 1 m/s). In this situation, the pristine ice crystals nucleated at the cloud top should have already undergone significant growth. Therefore, to discuss these cases with an open mind, we would like to mainly use $n_{ice,25um}$. We have carefully discussed all these possibilities in the revised manuscript.

**Comments 8:** Lines 308-312: If $S_i = 1.25$ in the 2nd case study as indicated above, then the cirrus cloud of Part A could be produced only by heterogeneous ice nucleation since $n_{ice,5um} = 635$ $L^{-1}$ and the mean dust-related INPC (U17-D) is 417 $L^{-1}$. This is contrary to what the article states here; that Part A is dominated by homogeneous ice nucleation.

**Response 8:** As strongly suggested by reviewer #2, here we decide to combine Part A and Part B and conduct the analysis by considering them as a whole. Therefore, the specific discussions have been largely updated and we hope that the reviewer can reevaluate our analysis.

**Comments 9:** Lines 350-352: Alternatively, could this also be explained by variability in cloud updraft velocities?

**Response 9:** We appreciate your emphasis on this valuable perspective. Variability in cloud updraft velocities has the potential to influence the generation and distribution of ice crystals within clouds. In further detail, small-scale vertical velocity perturbations can impact both the generation of ice crystals and the levels of ice supersaturation (Diao et al., 2015). Moreover, as demonstrated by Spichtinger and Gierens (2009), variations in updraft velocities within clouds can influence the competition between homogeneous and heterogeneous

nucleation. More enhanced updraft velocities can intensify homogeneous nucleation, leading to a greater number of small ice crystals; while ice crystals produced by heterogeneous nucleation take a less important portion in these regions. In the first case, the uneven distribution of $n_{ice,5um}$ should be attributed to the uncertainty in ICNC as suggested by reviewer #2. We have incorporated these insights by making revisions in the updated manuscript.

References:

Diao, M., Jensen, J. B., Pan, L. L., Homeyer, C. R., Honomichl, S., Bresch, J. F., and Bansemer, A.: Distributions of ice supersaturation and ice crystals from airborne observations in relation to upper tropospheric dynamical boundaries, J. Geophys. Res.-Atmos., 120, 5101–5121, https://doi.org/10.1002/2015JD023139, 2015.

Spichtinger, P., and Gierens, K. M.: Modelling of cirrus clouds–Part 2: Competition of different nucleation mechanisms, Atmos. Chem. Phys., 9, 2319-2334, https://doi.org/10.5194/acp-9-2319-2009, 2009.

**Comments 10:** Table 2: As mentioned earlier, for what purpose are ICNCs shown for $n_{ice,25um}$ and $n_{ice,100um}$? They provide no closure information regarding dust INPC.

**Response 10:** As previously mentioned in Response 5, we have decided to keep $n_{ice,25um}$ and $n_{ice,100um}$ based on our updated stratification for Case 1 and Case 2. This choice enables us to make more dependable comparisons between ICNC and INPC, especially when faced with considerable uncertainties in ICNC.

**Comments 11:** Lines 375-377: RHi (relative humidity with respect to ice) rarely reaches 140-150% in cirrus clouds since het (heterogeneous ice nucleation) always occurs before hom (homogeneous freezing nucleation), and INP and/or pre-existing ice tend to prevent the RHi from reaching the RHi threshold for hom. But if INP concentrations were low enough, the RHi threshold for hom would occur much more often to produce hom cirrus clouds, and their coverage could even exceed the coverage of het cirrus due to the smaller ice crystal sizes having lower fall speeds, as demonstrated in Mitchell et al. (2008, GRL). That is, lower ice sedimentation rates lead to longer cirrus lifetimes and greater cloud coverage. Moreover, the citation of Dekoutsidis et al. is misguided since that paper was showing hom is common in cirrus clouds and occurs mostly near cloud top where RHi is greatest, consistent with the modeling study by Spichtinger and Geirens (2009, ACP). The reference by Cziczo et al. does not support the author's claim either; rather it argues that most cirrus are het cirrus.

**Response 11:** Thanks for the constructive note suggested by the reviewer. As sufficient INPs are provided in this case, the situation in Mitchell et al. (2008) may not take place. According to the reviewer's suggestion, we have also removed the citations of Dekoutsidis et al. (2023) and Cziczo et al. (2013). Thus, we have fully rewritten this sentence as '**…Apart from homogeneous ice nucleation, the supply of efficient INPs contributes to an additional opportunity for the formation of cirrus clouds over open ocean regions; …**' (Please see lines 369-371)

**Comments 12:** Lines 377-380: While changes in UT INPC may alter the microphysical properties of cirrus clouds, this may not result in an increase in cloud cover and associated albedo for the reasons stated above. Moreover, the net radiative effect of cirrus clouds considers the absorption/emission of LW radiation in addition to SW radiation, and whether a net cooling or warming effect occurs may depend primarily on cloud optical thickness, the season, and the latitude.

**Response 12:** According to the reviewer's comments, we have rephrased the sentence as below '**the microphysical properties of cirrus clouds are modulated by differentiating ice-nucleating regimes, which may contribute different net radiation to the global climate.**' (Please see lines 371-372)

---

## Author Comment (AC2)

**Response to Reviewer #2**

**General Comments:**

The manuscript discusses two interesting dust-influenced cirrus events over the Pacific Ocean in the outflow regime of Eastern Asia. The remote-sending-based study makes use of spaceborne lidar and radar observations. The work is original and worthwhile to be published in ACP. The main effort is related to closure studies, i.e., deals with the question to what extent the ice-nucleation particle concentration (INPC, estimated from lidar observations) is in the same range of estimated ice crystal number concentration (ICNC, estimated from combined lidar-radar observations). Based on these ICNC-vs-INPC closure studies the authors also discuss to what extent homogeneous ice nucleation was involved in the (in situ) cirrus evolution.

Many aspects are unclear, a lot of speculative argumentation is given in the main result sections 3 and 4. A more careful discussion is requested. The uncertainty in all the retrieval products needs to be better considered. Therefore, major revisions are required.

One of the key points for me is: All the retrieval products (INPC, ICNC) can only be obtained within a large uncertainty range of an order of magnitude. ICNC cannot be obtained with an uncertainty of 25% as mentioned in the present study! Impossible! Even if you have well calibrated lidar and well calibrated cloud radar observations, and, in addition, Doppler information about fall speed of ice crystals (and thus shape and size information) the uncertainty can never be lower than expressed by a factor of 3-5 around the retrieval products. This is, e.g., shown in the reference, you provide (Ansmann et al., ACP, 2019). This is also discussed in detail by Buehl et al., AMT, 2019. Now, in the case of the CloudSAT radar we do not have Doppler information. So, the ICNC uncertainty is even higher. All in all, the uncertainty margins are roughly an order of magnitude for INPC and ICNC. That needs to be considered in the closure studies presented here and in all the conclusions drawn from the observations. To my opinion, this will make the full discussion easier and more straight forward.

**Response:** We appreciate the reviewer's thoughtful review and constructive comments. We have carefully checked the uncertainties in ICNC and changed the statement of '25% uncertainty in ICNC'. Accordingly, many revisions have been made in sections 3 and 4. All the comments have been addressed in the revised manuscript, and the responses to each comment are given below. We consider that this revised manuscript has been largely improved. The main revisions are listed below:

(1) As strongly suggested by reviewer #2, here we decide to combine Part A and Part B and conduct the analysis by considering them as a whole, due to the uncertainty of DARDAR retrievals. The specific discussions have been largely updated now.

(2) As at least three reviewers suggest shifting this manuscript from 'Measurement Report' to 'Research Article', we would like to do so but also need to involve Handling Editor Prof. Krämer in the decision. In addition, considering the results have been largely rewritten by adding in-depth discussions, we also think it would be better to change it to 'Research Article'.

(3) As the cirrus clouds in the two cases have already formed for at least half an hour (as deduced from the vertical extent of the ice virga by assuming a falling velocity of 1 m/s), nucleated ice crystals may have undergone significant growth. Therefore, in the comparison between ICNC and INPC, we decide to mainly use the values of $n_{ice,25um}$ and $n_{ice,100um}$. The specific reason has been discussed in the text of the revised manuscript.

(4) As there is no evident indication of the depletion of dust INP, the possibility for occurring homogeneous nucleation is low. Therefore, we have removed all the discussions about the involvement of homogeneous nucleation in the cirrus clouds.

**Detailed Comments:**

**Title:** This study is not a measurement report. This is an in-depth research approach (cirrus closure study)

based on the analysis of spaceborne observations. The term 'Measurement report' suggests that just robust observations (measured data) are presented. But the study is widely based on estimated products and interpretation of the findings.

**Response:** Taking the reviewer's suggestion, we have removed 'measurement report' in the title. Here we would also like to involve our Handling Editor Prof. Krämer to judge if it is justified to shift this manuscript from 'Measurement Report' to 'Research Article.'

**Abstract:** The abstract needs to be rewritten after all necessary changes.

**Response:** We have modified the abstract according to the revisions in the text.

**Introduction:**

I would keep the introduction as short as possible. There are so many cirrus papers and all present lengthy paragraphs on the radiative impact. I would suggest that the importance of cirrus in the atmospheric system is just briefly described, followed by gaps in our knowledge regarding cirrus evolution, and then what you are going to present in this study.

**Response:** Considering the reviewer's suggestion, we have shortened the start part of the introduction so that we can quickly enter into the research gap in the field and our research topic. The detailed contents regarding the radiative impact have been removed. In the revised manuscript, the first and second (just the beginning part) paragraphs have been rewritten. (please see lines 35-52)

Lines 41-42: Note that homogeneous freezing also needs aerosol particles, however pure liquid ones, such as sulfate aerosol (without any insoluble part). In the case of heterogeneous ice nucleation, one needs particles with an insoluble fraction (sites for ice nucleation).

**Response:** We have replaced '(spontaneous)' with '(e.g., sulfate aerosol solution)'.

Figure1: Does MERRA also deliver dust profiles? or only column mass values? Would be nice to have model dust profiles up to cirrus level.

**Response:** Thank you for the suggestion. The MERRA-2 data only provide column dust mass density. For dust situation at cirrus levels, here we utilize CAMS global reanalysis (EAC4) data to plot the daily-mean size-resolved dust aerosol mixing ratios on 2-5 May 2010 (as shown in the figure below). However, it does not trace the transport pathway of the dust plumes. Therefore, if possible, we would like to retain the use of MERRA-2 column dust properties to indicate that Asian dust may indeed be transported to the central of (or even across) the Pacific. In the case studies later, we will show the height distributions of the dust layers with CALIOP observations.

[Figure]

**Method section:**

Line 130: 25% uncertainty is unrealistic as discussed above. To use uncertainty margins of an order of magnitude (a factor of 3 around the retrieved value) makes sense. That is more realistic!

**Response:** Thank you very much for pointing out this issue. We have modified the uncertainty in ICNC to be a factor of 3 (Sourdeval et al., 2018) in section 2.2 as well as the rest part of the manuscript. (Please see lines 125-137)

Lines 140-145: I would remove all immersion freezing parameterizations. The two case studies deal with cirrus from 9 to 11.2 km height, and temperatures from -40 to -54°C. Furthermore, your Table 2 considers deposition ice nucleation, only (no immersion freezing INP values are given). Even in virga zones with higher temperatures, there is ice saturation or even ice sub saturation (and especially sub saturation with respect to water). Immersion freezing is impossible in the presence of in situ cirrus with well-developed virga structures.

**Response:** We have removed the immersion freezing from the text as well as from figures 6 and 10. Now the statements here are taken as '**Considering the temperatures at cirrus levels, we only considered deposition freezing in cirrus clouds (Kanji et al., 2017; Marcolli, 2014) by applying the parameterization scheme U17-D (Ullrich et al., 2017) for dust-related INPC computation. Here we disregarded contact freezing as well as immersion freezing, which needs an INP to collide with/immerse in a supercooled droplet.**' (please see lines 148-151)

Table 1: Are all the equations needed? A list of parameters and retrieval uncertainties makes sense. Again, please skip all immersion freezing parameterizations.

**Response:** We have removed the immersion freezing parameterizations in Table 1. As for the rest equations listed, if possible, we would like to retain them so that the readers without the background of the POLIPHON method can easily follow the work.

**Result section:**

The cirrus shown in Figure 3 is a classical in-situ cirrus, with a well-defined cirrus top region, obviously above 10 km height, and a virga zone from 10 km down to 6 km height. The cirrus is clearly dust influenced.

Let me explain how such cirrus layers develop: Ice nucleation starts at cloud top, at the coldest point of the cloud, here the ice nucleation probability is highest. And if dust particles are present, they will trigger ice

nucleation. Then diffusional growth of the crystals takes place, collision and aggregation. Sedimentation of ice crystals begins and the growth of the crystals leads to an immediate reduction in relative humidity (throughout the cirrus layer from 6 to 10.5 km height). There is almost no room for homogeneous ice nucleation (in the case of a well-developed cirrus system) because there is practically no potential to create a scenario with sufficiently high relative humidity over ice with values exceeding 150%, required for homogeneous ice nucleation. Homogeneous freezing is only possible at cloud top in the case of rather strong updrafts with large updraft speed so that super saturations levels develop faster than INPs can be activated (to reduce super saturation). All this is described in Kaercher et al., JGR, 2022. However, such a scenario with rather strong updrafts is not visible in Figure 3. A very harmonic cirrus development exclusively controlled by the activation of dust INPs seems to be realistic. Furthermore, radiative cooling at cloud top of a well-developed cirrus system will cause some amount of downward motion and may contribute to a suppression of such rather strong updrafts. If we keep an uncertainty of about an order of magnitude into account for ICNC and INPC, all observations support that heterogeneous ice nucleation on dust particles dominated. Because of the large uncertainty, I would not further analyze the defined observational periods (part A and part B), i.e., explain the differences in the ICNC values in terms of heterogeneous vs homogeneous ice nucleation. The differences in the results for part A and B just show, to my opinion, the uncertainty in the products.

**Response:** We are grateful for the reviewer's valuable comments and agree with the opinion that the possibility for homogeneous nucleation in this cirrus is rather small. Thus, we have revised the analysis area of the cirrus cloud by combining Part A and Part B in each case and explain the high ICNC values by the updated uncertainty in ICNC (a factor of 3 as also mentioned above). The subsequent calculations and analysis are all based on this update.

**Back to the study:**
Figure 3: The lidar is able to see the small ice crystals after nucleation with sizes of 1-5 micrometers at cloud top, above 10 km. The cloud radar seems to be not able to detect the ice crystals above 10 km height (Figure 5). The radar detects the ice crystals after diffusional growth and collision and aggregation processes, and after the start of sedimentation processes, i.e., several 100 m below cloud top, in the virga zone, as can be seen in Figure 5a and 5c. So, lidar-radar retrievals will not be able to exactly see the ice crystal number concentration of freshly nucleated ice crystals at cirrus top. And when radar comes into plays, aggregation took already place, and the number of crystals already decreased, may be already by a factor of 2, or even a factor of 3-5. Another point: Can we assume a 'classical' size distribution (as typically measured with aircraft instrumentation) in the case of freshly formed ice crystals? The size distribution is input in the DARDAR approach, and may be very narrow for the freshly nucleated crystals? All these unknown aspects cause the large uncertainty in the ICNC products (of at least one order of magnitude).

**Response:** Thank you for your valuable comments. Indeed, lidar is more sensitive to the small particles near the cloud top and radar is likely to be sensitive to the large ice particles that have already fallen for a while with the probable experience of collision and aggregation processes. For the DARDAR Nice data product, Sourdeval et al. (2018) mentioned that it is difficult to quantify the overall uncertainties in ICNC caused by instrumental sensitivity and physical assumptions (e.g., parameterization scheme for the scattering properties of ice crystals, particle size distribution of ice crystals, and lidar ratio for ice crystal). They instead evaluated the quality of ICNC by comparing them with in situ measurement and concluded that the PDS assumption contributes to the dominant error in ICNC (as seen in section 4.2 therein). It should be mentioned that for in situ measurements of ICNC, the two-dimension stereo (2D-S) probe for particle sizes of 5-1280 μm can be associated with large uncertainties in the first two bins (5-25 μm) as suffering from uncertainties due to instrumental response time and depth of field (Gurganus and Lawson, 2018) as well as the shattering effect (Korolev et al., 2015). Overall, in Sourdeval et al. (2018) about a factor of 2 overestimation was stated as the uncertainty in $n_{ice,5um}$ and $n_{ice,25um}$ due to a misrepresentation of the PSD shape by Delanoë et al. (2005) at warm temperatures. In this study, we conservatively consider the uncertainties in $n_{ice,5um}$ and $n_{ice,25um}$ to be a factor of 3-5 for the selected cases with both lidar and radar available. As for $n_{ice,100um}$, assumed PSD shape by

Delanoë et al. (2005), in principle, performs best at reproducing the concentration in large size, and in situ measurements also show better accuracy at this size range. We have added the related sentences in the second paragraph of section 2.2. (Please see lines 125-137)

**References:**

Delanoë, J., Protat, A., Testud, J., Bouniol, D., Heymsfield, A. J., Bansemer, A., Brown, P. R. A., and Forbes, R. M.: Statistical properties of the normalized ice particle size distribution, J. Geophys. Res.-Atmos., 110, D10201, https://doi.org/10.1029/2004JD005405, 2005.

Gurganus, C. and Lawson, P.: Laboratory and Flight Tests of 2D Imaging Probes: Toward a Better Understanding of Instrument Performance and the Impact on Archived Data, J. Atmos. Ocean. Tech., 35, 1533–1553, https://doi.org/10.1175/JTECHD-17-0202.1, 2018.

Sourdeval, O., Gryspeerdt, E., Krämer, M., Goren, T., Delanoë, J., Afchine, A., Hemmer, F., and Quaas, J.: Ice crystal number concentration estimates from lidar–radar satellite remote sensing – Part 1: Method and evaluation, Atmos. Chem. Phys., 18, 14327–14350, https://doi.org/10.5194/acp-18-14327-2018, 2018.

Korolev, A. and Field, P. R.: Assessment of the performance of the inter-arrival time algorithm to identify ice shattering artifacts in cloud particle probe measurements, Atmos. Meas. Tech., 8, 761–777, https://doi.org/10.5194/amt-8-761-2015, 2015.

The ICNC values in Figure 5d vary strongly and indicate the uncertainty in all the retrieval products. Therefore, I would not introduce part A and part B and 'believe' that the rather different findings are caused by different ice nucleation processes. One may formulate hypotheses…, but one needs to consider the large uncertainty in the discussion. Solid conclusions are difficult to draw. And as I mentioned, I am skeptical that homogeneous ice nucleation has a chance to occur in the presence of a well-developed cirrus. The CALIOP lidar indicates dust around the cirrus and no indication that the dust INP reservoir was depleted. To my opinion, a depletion of the INP reservoir is unlikely during part A and B, and therefore homogeneous ice nucleation is unlikely.

**Response:** Considering the reviewer's suggestion, we have combined Part A and Part B and analyzed them as a whole. We agree with the reviewer that it is probably a heterogeneous nucleation case since it is a well-developed cirrus and the INP supply seems rather sufficient. Therefore, we have added some related sentences in the second paragraph of section 3.1 accordingly. (please see lines 215-217)

If we keep the uncertainty of one order of magnitude in mind, the ICNC values for part A and B shown in Figure 6e nicely indicate the ICNC uncertainty range. To repeat, to my opinion, only heterogeneous ice nucleation makes sense. In the presence of so many rather favorite dust INP particles (as shown in Fig. 6c, 4000 large dust particles with a diameter > 500 nm per liter were present, and the corresponding dust surface area concentration was high with values up to 40 $\mu m^2$ $cm^{-3}$) homogeneous ice nucleation is rather unlikely.

**Response:** Considering the reviewer's suggestion, we have combined Part A and Part B and analyzed them as a whole. The related statements have been modified now. (Please see lines 228-233, 252-262)

Figure 6 indicates that INPC values for S-ice =1.15-1.25 (Ullrich parameterization, deposition ice nucleation) seem to be very likely (or reasonable) and match roughly the ICNC values (n-ice for crystals with sizes > 5 μm, for the periods of part A and B), when keeping in mind that ICNC is probably underestimated close to the cirrus top (because DARDAR values are not very trustworthy here because of too weak or even missing radar reflectivity values). The best DARDAR values are shown after significant growth of crystals and after aggregation processes (from 9-9.6 km height), but then the ICNC values are already reduced compared to the number of freshly nucleated crystals at cirrus top, probably reduced by a factor of 2-5.

**Response:** We are grateful for the comments that are rather helpful for explaining the data and analyzing the physical process. We have added some sentences to discuss this issue as below '**… It should be also mentioned that ICNC values near the cloud top may be more uncertain because of the weak or missing radar reflectivity, which is not sensitive to the small ice crystals here…. and then undergo aggregation process with a reduction of ICNC by a factor of 3-10 (Field and Heymsfield, 2003)….**' (please see lines

Line 242: As mentioned above, homogeneous freezing in an environment with already existing ice crystals (and thus super saturation values S-ice around 1.0) is very unlikely. When keeping the high dust load and the large uncertainty margins into account, the closure is fine, ICNC and INPC match reasonably well. This is ok! This is a good result.

**Response:** We have rewritten the related sentences in this paragraph. (Please see lines 252-262)

In summary here, please, do not compare part A and part B, just take the average of both periods and use the averaged values for comparison with the Ullrich results for INPC. Ice super saturation values of 1.15 are close to the values in the paper of Ansmann et al. (2019) for pure (unpolluted) dust scenarios. In the case of aged dust (or polluted dust, case study 2 in this study here) super saturations of 1.35 make sense. Such values are assumed to be realistic for aged, coated dust particles (see Kaercher et al, JGR, 2022). Homogeneous ice nucleation events do not make sense to me at all. However, I leave it open to you to find a proper and careful argumentation for the potential contribution of homogenous ice nucleation.

**Response:** We have combined Part A and Part B and analyzed them as a whole. Here we find good agreement between U17-D-derived INPCs with Si of 1.25 and $n_{ice,5\mu m}$. We have removed the discussions about the possibility of homogeneous nucleation here.

In the discussions (sections 3 and 4) there are many speculative aspects. Speculations are not justified. A more careful interpretation of the results is needed. And if you have a hypothesis, start with: We hypothesize…. and then the hypothesis must be based on convincing argumentation. And please keep the large uncertainties in mind.

**Response:** We have rephrased many sentences and made significant modifications to these two sections to avoid the hypothesis of homogeneous nucleation, especially considering that the INP supply here is rather sufficient.

**Case study 2:**
Another nice case with an impact of dust. In this case, CALIOP aerosol typing seem to indicated polluted or coated dust. The ice nucleation efficiency of aged and coated dust particles may be reduced by a factor of 5-10 compared to the ice activity for pure dust. This may or could be considered when computing INPC values with the Ullrich parameterization by multiplying the Ullrich INP values by 0.1-0.2.

**Response:** We have added a sentence to mention that the dust particles included in the dust layer are mainly aged/polluted. Thanks to the reviewer's comments, we have also updated the analysis/discussion part of this case by multiplying the U17-derived INPCs by a factor of 0.1. (Please see lines 275-276, 312-322)

Again, I would not compare the results for part A and part B because of the large uncertainties. The results in Figure 10e support my comment here. The results as a whole (part A and B) are in good agreement with the Ullrich INP values for S-ice of 1.35 when considering a factor of 0.1-0.2 less INPC (in the case of polluted or coated, less ice nucleation efficient dust). Kaercher et al., JGR, 2022, used S-ice values around 1.35 for the activation threshold for polluted dust. In case 2, the ICNC values (from the DARDAR approach) increase up to cloud top. Obviously, the radar reflectivity values were strong enough to obtain reasonable ICNC values even close to the nucleation range at cloud top.

**Response:** We have combined Part A and Part B and analyzed them as a whole. Considering the 'modified' U17-D-derived INPC values by multiplying a factor of 0.1, good agreement between INPC and $n_{ice,25\mu m}$, (or $n_{ice,100\mu m}$) can be seen. The related statements have been added. (Please see lines 319-323)

To my opinion, there is again no room for homogeneous ice nucleation. There is dust 'before' and 'after' the cloud region, so a depletion of the dust INP reservoir is not visible. And at these conditions, homogeneous ice

nucleation is unlikely.

**Response:** In the last paragraph of section 3.2, we have stated that this case is probably dominated by heterogeneous nucleation. (Please see lines 323-324)

Please avoid speculations on cirrus type, etc…. in sections 3 and 4. Just mention, what is really available from the observations.

**Response:** We have removed the speculations on the cirrus type.

**Discussion section:**

The first paragraph is not needed to my opinion.

**Response:** We have removed the first paragraph of the discussion section.

Line 346: Can we have longitude-latitude information (not only latitude). How long (in km) was the cirrus layer? The same for case study 1.

**Response:** The longitude ranges have been added to the text. We have also added the horizontal extent of the cirrus clouds in these two cases in section 3.3. (Please see lines 339 and 345)

Be careful with 'dominating homogeneous freezing' in environments with so much dust. It is simply difficult to produce high ice super saturation in the presence of favorable INPs.

**Response:** We have removed the statements regarding the possibility of homogeneous nucleation.

If you follow my suggestions, you can significantly 'improve' Table 2 by reducing the information content. I think ICNC for n-ice (> 5 µm) has to be compared to INPC, however information on n-25, n-100 is useful as well, especially to get a better feeling for the large uncertainties in all ICNC products.

**Response:** We have combined the Part A and Part B and updated the Table 2. And $n_{ice,25\mu m}$ and $n_{ice,100\mu m}$ are retained in the table as suggested by the reviewer.

In the case of the Ullrich INPC values, input is dust surface area concentration, S-ice as well as temperature! Temperature needs to be mentioned in Table 2 (Ullrich INP values).

**Response:** Thanks for your reminder. Temperatures (mean value as well as the maximum and minimum) for each case have been added in Table 2.

---

## Author Comment (AC3)

**Response to Reviewer #3**

**Referee report for the manuscript entitled:**

"Measurement report: Influence of long-range transported dust on cirrus cloud formation over remote ocean: Case studies near Midway Island, Pacific"

Authored by Huijia Shen, Zhenping Yin, Yun He, Longlong Wang, Yifan Zhan, Dongzhe Jing

In this manuscript, remote sensing data has been used to determine whether long-range transported dust from Asia is active as INP in the formation of cirrus clouds. Data are collated from two space-borne instruments to calculate INP concentrations (INPC) and ice crystal number concentrations (ICNC). These values are compared at specific locations within an observed cirrus cloud that relate to where the dust would most likely be entrained. Where the values of INPC and ICNC are comparable, the manuscript concludes that this region is dominated by heterogeneous freezing due to the dust. Where the ICNC is much higher than the INPC, the manuscripts concludes that homogeneous freezing also takes place. Previous studies have shown that long-range transported dust acts as INP (Saharan dust in North America) and Asian dust has been shown to act as INP in China. This manuscript shows that Asian long-range transported dust also acts as INP. I am not an expert of INP, however having experience in adjacent aerosol fields, I am aware that there is still much to understand about INP sources and this manuscript confirms another significant source of dust and its far-reaching impacts.

When considering this manuscript for publication, there are several factors to take into account:

This manuscript is presented as a measurement report, however the data are retrieved from space-borne instruments rather than field or lab work. Additionally, there is analysis and interpretation of the results which potentially goes beyond the requirements of a measurement report.

This manuscript does not present novel ideas or methods, however it does establish evidence of long-range transported dust acting as INP in a new area.

For me, there are some major questions about the analysis of the individual boxes within the cloud. It appears that one of the boxes closest to the dust plume is described as lacking in INP and, as such, homogeneous freezing is dominant. This description requires some justification.

**Response:** We are grateful for the reviewer's constructive comments on the manuscript. All the comments have been addressed in the revised manuscript, and the responses to each comment are given below. Therefore, the specific discussions have been largely updated now. The main revisions are listed below:

(1) As strongly suggested by reviewer #2, here we decide to combine Part A and Part B and conduct the analysis by considering them as a whole, due to the uncertainty of DARDAR retrievals. The specific discussions have been largely updated now.

(2) As at least three reviewers suggest shifting this manuscript from 'Measurement Report' to 'Research Article', we would like to do so but also need to involve Handling Editor Prof. Krämer in the decision. In addition, considering the results have been largely rewritten by adding in-depth discussions, we also think it would be better to change it to 'Research Article'.

(3) As the cirrus clouds in the two cases have already formed for at least half an hour (as deduced from the vertical extent of the ice virga by assuming a falling velocity of 1 m/s), nucleated ice crystals may have undergone significant growth. Therefore, in the comparison between ICNC and INPC, we decide to mainly use the values of $n_{ice,25um}$ and $n_{ice,100um}$. The specific reason has been discussed in the text of the revised manuscript.

(4) As there is no evident indication of the depletion of dust INP, the possibility for occurring homogeneous nucleation is low. Therefore, we have removed all the discussions about the involvement of homogeneous nucleation in the cirrus clouds.

The methods would be enhanced by some more detailed description.

**Response:** Combining the comments from the four reviewers, we have realized that the uncertainties in DARDAR ICNCs are the most important concern that should be emphasized. Therefore, we have added a description of this issue in the second paragraph of section 2.2. (please see lines 125-137)

Restructuring of some paragraphs and tightening up the language in a few key sentences would greatly aid the reading and understanding of the manuscript.

Based on the above summary and the comments below, I recommend this manuscript is reconsidered after major revisions. I also recommend that this should be submitted as a research article if the following suggestions make the manuscript a more substantial contribution. Note that I do not have expertise in satellite products so am unable to comment on their descriptions here and the suitability of their use.

**Response:** Thank you for the valuable comments. We have largely revised the manuscript by considering the comments from you and the other three reviewers. Taking the reviewer's suggestion, we have removed 'Measurement report' in the title. Here we would also like to involve our Handling Editor Prof. Krämer to judge if it is justified to shift this manuscript from 'Measurement Report' to 'Research Article.'

**Major points:**

On the description of the method, the key aspects from He et al. (2022b) are mentioned throughout the manuscript, but it is quite scattered. I would suggest pulling this all together within the methods section so that the method is very clear for a reader who is unfamiliar with it. It is clear that if the ICNC is much higher than INPC then homogeneous freezing must be present, but it is not so clear how one knows that it is only heterogeneous if the ICNC is comparable with INPC. Why could it not still be both? I am not an expert in INP, but I think the key bit of information here is that the ice saturation will not reach the required 140 or 150% if heterogeneous freezing has already begun.

**Response:** Thank you for the reviewer's constructive comments. The details of the methodology were given in He et al. (2022). For this study, the significance is to reveal the potential influence of the long-range transoceanic Asian dust on the ice formation of cirrus clouds over the central Pacific, where is usually clean. Therefore, the methodology was not thoroughly described to avoid redundancy. It should be pointed out that, in either He et al. (2022b) or this study, the proposed method is only applied to case studies. One should note that the current space-borne remote sensing approach can help to find out only the dominant ice nucleation regime or the other due to the limited spatiotemporal resolution, insufficient sensitivity of radar signal at cloud top, and the relatively large retrieval uncertainty. Therefore, it may still count on in-situ measurements (Krämer et al., 2016, 2020; Sourdeval et al., 2018). The uncertainty in ICNC from the DARDAR product is estimated to be a factor of 3 due to instrumental sensitivity and physical assumptions; the uncertainty in INPC can be a factor of 0.5-5 due to the retrieval of dust-related INP-relevant parameters and limited accuracy of INP parameterizations. Besides, we also have to adjust the values of ice saturation and modified factor (polluted/aged dust will have weaker ice-nucleating efficiency) in INP calculations in different cases. As a result, we consider an ICNC-to-INPC ratio within one order of magnitude to be a good agreement.

For the second issue, if the ICNC is comparable with INPC, the most possible situation is that heterogeneous nucleation suppresses homogeneous nucleation by consuming the water vapor (i.e., reducing the ice saturation). This process can be continuous until the depletion of INP. In addition, there is another possibility that a very strong updraft is present so that homogeneous nucleation has a chance to immediately take place before the involvement of heterogeneous nucleation; in this situation, heterogeneous nucleation will finally occur later considering the sufficient INP supply. All in all, the actual physical process is rather complicated, the different types of ice nucleation mechanisms can take place one after another or simultaneously for a short while (i.e., the transition phase between the two mechanisms).

**References:**

Krämer, M., Rolf, C., Luebke, A., Afchine, A., Spelten, N., Costa, A., Meyer, J., Zöger, M., Smith, J., Herman, R. L., Buchholz, B., Ebert, V., Baumgardner, D., Borrmann, S., Klingebiel, M., and Avallone, L.: A microphysics guide to cirrus clouds Part 1: Cirrus types, Atmos. Chem. Phys., 16, 3463–3483, https://doi.org/10.5194/acp-16-3463-2016, 2016.

Krämer, M., Rolf, C., Spelten, N., Afchine, A., Fahey, D., Jensen, E., Khaykin, S., Kuhn, T., Lawson, P., Lykov, A., Pan, L. L., Riese, M., Rollins, A., Stroh, F., Thornberry, T., Wolf, V., Woods, S., Spichtinger, P., Quaas, J., and Sourdeval, O.: A microphysics guide to cirrus – Part 2: Climatologies of clouds and humidity from observations, Atmos. Chem. Phys., 20, 12569–12608, https://doi.org/10.5194/acp-20-12569-2020, 2020.

Sourdeval, O., Gryspeerdt, E., Krämer, M., Goren, T., Delanoë, J., Afchine, A., Hemmer, F., and Quaas, J.: Ice

crystal number concentration estimates from lidar–radar satellite remote sensing – Part 1: Method and evaluation, Atmos. Chem. Phys., 18, 14327–14350, https://doi.org/10.5194/acp-18-14327-2018, 2018.

The methods could also contain a description of the two case studies before we dive into the results. In the discussion there is a nice point about the dust being from intense events where the dust is elevated by Mongolian anticyclones. Information like this would be better used in the beginning of the manuscript to set the scene. Is there other relevant meteorological information about the cases? Was there a particular reason for picking these two cases? What times and altitudes were used? Are the cirrus events very rare? A description of the two case studies would follow on nicely from the HYSPLIT model description in Section 2.

**Response:** There is some background information about the initiation of the Asian dust event (Sun et al., 2001). Asian dust plumes from the Gobi are always associated with the Mongolian anticyclone cases in which strong surface wind can blow up the dust particles and lift them to the troposphere, which then may be advectively transported. For the Asian dust plumes from the Taklimakan Desert, dust particles are elevated via the upward winds caused by the convergence of warm and cold air systems but most lifted dust below 5 km cannot be involved in the long-range transport due to the terrain confine from the mountains surrounding (i.e., West, North, and South sides). However, here we more focus on the dust-cirrus interactions rather than the initial mechanism of the Asian dust events.

It should be mentioned that it is not easy to find such clear-cut cases in which dust particles can be seen in the vicinity of (to connect with) the cirrus clouds. The cases presented in this manuscript can lead to a deeper analysis. And the dataset of collocated measurements from CALIPSO and CloudSat is limited. Hence, we do not conduct a statistical study. The launch of EarthCARE (with a 94 GHz Doppler cloud radar and a high spectral resolution lidar) in 2024 may enrich the measurements and provide many more cases for such a statistical study (Wehr et al., 2023).

**References:**
Wehr, T., Kubota, T., Tzeremes, G., Wallace, K., Nakatsuka, H., Ohno, Y., Koopman, R., Rusli, S., Kikuchi, M., Eisinger, M., Tanaka, T., Taga, M., Deghaye, P., Tomita, E., and Bernaerts, D.: The EarthCARE mission – science and system overview, Atmos. Meas. Tech., 16, 3581–3608, https://doi.org/10.5194/amt-16-3581-2023, 2023.

Sun, J., Zhang, M., and Liu, T.: Spatial and temporal characteristics of dust storms in China and its surrounding regions, 1960–1999: Relations to source area and climate, J. Geophys. Res., 106, D10, 10325–10333, https://doi.org/10.1029/2000JD900665, 2001.

The results could be restructured to make the storyline clearer. For each case, I would recommend starting with establishing the case using the HYSPLIT trajectory and the dust mass column density plots. Then the description of the cloud and the dust plume using the CALIPSO cross-sections would follow on nicely. Next, part A and part B need a clear introduction using the DARDAR product. Why were these regions in particular picked? The difference in the ICNC between the two boxes could be described and then finally use the profiles to compare the ICNC values with the INPC. Here, I would suggest stating the ICNC versus the INPC and the ratio for part A and B. Do these values alone indicate homogeneous or heterogeneous freezing? The main method used in the manuscript is this quantification so I would pull it out of the data and emphasise it. Then add the additional, contextual information from other papers about typical values for each mechanism.

**Response:** We are grateful for the suggestion. Just let us explain why we organize each case study as it is shown now. In this study, the most important thing is that CALIOP offers real observation to see that cirrus and dust are both there and have connected. Only with this fact, one can then use the MERRA-2 analysis data and HYSPLIT model as auxiliaries to trace/confirm the possible source and transport pathway of the air mass (i.e., dust plume). As MERRA-2 only shows the column information about the dust, we cannot use it to conjecture whether there are dust plumes at cirrus altitudes; also, no information on the presence of cirrus clouds can be provided with both MERRA-2 and HYSPLIT model. Therefore, it would be better to retain the structure as is and we hope the understanding from the reviewer.

As for the discussions of Part A and Part B, as suggested by reviewer #2, we have to combine them as a whole for further analysis. Thus, the result part has been largely modified. The reviewer may have to reevaluate it.

Table 2 shows the summary of ICNC and INPC for both cases split into the A and B boxes. Considering that the concentrations are defined as being "comparable" as within an order of magnitude, there does not seem to

be a large distinction between the A and B boxes. This is particularly true for the first case where the ICNC values are quite similar for both boxes. For the second case, there does indeed seem to be a distinction between the boxes, however the INPC values still seem much higher than both boxes. These values would benefit from some clarification, and perhaps this could come in the results section if it were restructured as recommended above.

**Response:** Thank you for your valuable comments. According to the reviewer's suggestion as well as the comments from reviewer #2, we have made significant modifications including merging Part A and Part B in each case, considering that the two parts merely reflected differences in retrieval uncertainty without substantial distinctions. The revised ICNC-INPC comparative results and analysis will be presented based on these modifications.

**Minor points:**

The title could state the outcome of the study since an effect has been found by this study. For example, "Long-range transported Asian dust plumes influence cirrus formation over remote ocean."

**Response:** We have changed the title a bit to **'Long-range transported Asian dust plumes influence cirrus cloud formation over remote ocean: Case studies near Midway Island, Pacific'.**

The abstract has a nice structure to it, but the sentence beginning "However, it is still not well…" (line 14) is confusing. This is a key sentence and rephrasing it would make the motivation of the research clear in the abstract. The results could also be stated more clearly, for example including the ICNC - INPC ratio.

**Response:** We have rephrased the entire abstract in the revised manuscript.

The comment on line 47 about geoengineering comes across as quite random and a bit of an afterthought. Perhaps it could be rephrased to tie in with the previous sentence about the uncertainty and how understanding of these clouds needs to be improved in order for cloud seeding to be done appropriately.

**Response:** Thank you for pointing out this. We have been requested to shorten the introduction by another reviewer so as to quickly enter into the main topics of this study. After consideration, we decide to remove the statements related to geoengineering and cloud seeding.

The introduction has good content to give context to and justify the study. It does very well to lay out why we care about cirrus and this long-range dust transport. However, a couple of paragraphs here could be restructured to improve the logic:

**Response:** We have made some revisions according to the reviewer's suggestions.

The paragraph beginning at line 50 could be split into first discussing the pristine environment, low AOD, and the related lack of observations of cirrus. It is also a bit unclear whether there are not many cirrus or there are many but it is unexpected because of the pristine environment. Then the question of whether this means cirrus are purely formed by homogeneous could be framed.

**Response:** We have revised the manuscript to more clearly state that there are many (occurrence of 0.3-0.4) but it is unexpected because of the pristine environment. (please see lines 58-60)

The ice saturation conditions for both mechanisms are a key part of the background science and this could be written in a separate paragraph. This could then link dust as an INP and give a description of why homogeneous freezing is much less likely to occur if dust is present (high ice saturation is prevented).

**Response:** We are grateful for the reviewer's suggestions about the organization of this paragraph. We have separated the different requirements of ice saturation conditions for heterogenous and homogeneous ice nucleation as a single paragraph (now the second paragraph of section 1), and have also made some adjustments. (please see lines 45-48)

As for the second issue, we consider, if possible, detailed physics such as '(high ice saturation is prevented in the presence of dust' would be better to provide in the results and discussions.

The paragraph beginning at line 67 could also be restructured. From the studies cited here, I have understood that we know Saharan dust travels long distance, we know that Asian dust affects cirrus in China and also travels long distance to North America. So it seems to me that the question is not really whether the dust affects cirrus a long distance away, but rather is this long-distance transport creating a significant effect in an

otherwise pristine environment? This paragraph could lead the reader more to this question about what is happening over the ocean.

**Response:** Thank you for the suggestion of the organization of this paragraph. Please also see the second sentence of this paragraph which is read as '*Hence, the possible influence of transoceanic dust particles on cirrus cloud formation should be considered.*' As requested by the reviewer, this sentence has raised a question about connecting the long-range transport dust to cirrus clouds. Afterward, we begin to introduce several very relevant studies regarding the influence of transoceanic dust on cirrus formation. Then, the topic of dust-cirrus interaction at the halfway of transpacific transport is raised at the end of this paragraph. Thus, we consider it would have already followed the reviewer's logic to structure the paragraph.

The paragraph beginning at line 83 could start with stating that He et al (2022b) determined the freezing mechanism by comparing INPC and ICNC before explaining why this works. Some clarification explaining how one knows that a mechanism is "dominant", rather than there just being a mixture of both, would help the reader. There is good placement of the descriptions of the products here and nice summary with referencing.

**Response:** For this method, we can only consider the possibility of '(1) sole heterogeneous nucleation' or '(2) homogeneous nucleation also involves (i.e., it can be considered as the mixture situation as the reviewer mentioned)'. This means one cannot judge whether there is a possibility for sole homogeneous nucleation. Because the actual cloud physical process is much more complicated. For (1), it is easy to determine. However, for (2), it may be more complicated. If dust INPs are provided, heterogenous nucleation can take place first, and then, along with the depletion of INP homogeneous nucleation may begin to be involved. Or, if a very strong updraft is present, homogeneous nucleation can rapidly take place regardless of a sufficient supply of dust INP, because there is not enough time for triggering heterogeneous nucleation. Considering the complexity, we would like to avoid discussing all of these details in the introduction and hope that the reviewer can understand.

The point about seeder feeder in stratocumulus clouds (line 80) could be made earlier because it is addressing the big question of why do we care about these cirrus? Perhaps it could be stated near the geoengineering comment as wider motivation for understanding cirrus.

**Response:** As requested by another reviewer, we have to shorten the introduction part. The statements regarding the geoengineering and seeder-feeder process have been removed now.

On line 82, there is the phrase "it is indispensable" but it is unclear what it is. I am not sure if this line adds anything.

**Response:** We have removed this sentence.

The overview of the paper (line 103) could be clearer and include "section 3".

**Response:** We have revised.

The manuscript tends to state what the figure is in the text, e.g. "Figure 5 shows the ice cloud properties, including cloud extinction coefficient, cloud particle effective radius, ice water content, 200 and ice crystal (with size <5 μm)....". They could consider removing this from the main text since it is already described in the figure caption. This may make the story flow better as the main text can then go straight into what is observed in the figure, e.g. "Figure 5b shows the region of dust-related cirrus... ".

**Response:** We have made the necessary adjustments in the manuscript to avoid this issue.

The introduction to the results sectioned could be cut as there is some repetition that one INP is generating one ice crystal and that secondary ice productions is not being considered to take place in these conditions. But the statement of what is required for "good" agreement and the definition of dust-cirrus interaction event are well placed here.

**Response:** We have made corresponding revisions for clarity. (Please see lines 180-182)

The authors might consider adding an explanation for why the description of the dust in the main text based on figure 3a and b (line 184) does not fit with where the dust is identified in figure 3c and e.

**Response:** In figures 3a, b, and d, we mark the same region to write the words beside 'Dust-related cirrus'. For figure 3c, since it is the vertical feature mask that can identify aerosol and cloud, we use a different

rectangle to include the whole region of conterminous cirrus and dust and write the words beside 'dust-cirrus interaction'. For figure 3e, only aerosol subtypes are shown; thus, we just mark the region of dust aerosols and write the words beside 'Dust INP'. All in all, these different rectangles just aim to roughly indicate the regions where dust-cirrus interaction takes place rather than to show the specific regions employed for further quantitative study/analysis. The specific regions of cirrus and dust for analysis are given later in figure 6 as well as in Table 2. We have added a sentence to remind this in the first paragraph of section 3.1. (Please see lines 193-195)

In the main text about figure 5, the in-cloud averages are stated but there is no interpretation of these. In the introduction it is stated that homogeneous freezing produces more, smaller ice crystals. Do the in-box averages support the allocation of homogeneous/heterogeneous freezing based on the radius and ice water content?
**Response:** Indeed, extinction coefficient, effective radius, and ice water content can be considered the input parameters for calculating the ICNC in the DARDAR Nice product. However, it is difficult to just use these parameters to discuss which type of ice nucleation mechanism should take responsibility for the ice formation. Therefore, we only intend to provide all the available information in this case study. Detailed discussion regarding the ice nucleation mechanism will be left to the ICNC-INPC comparison part.

Some clarification on line 236 about "pristine ice crystals" in abundant INP near the top of the cloud could help the reader. Are these pristine ice crystals because they are from homogeneous freezing or because they are formed in pristine environments, where the INP are more numerous at cloud top? Perhaps changing the description of the INP from abundant to more numerous would help.
**Response:** The 'pristine' just means 'initial-formed' ice crystals formed near the cloud top regardless of the type of ice nucleation (usually considered within the range of 350 m below the cloud top, Bühl et al., 2016) and without further physical process, e.g., significant growth, aggregation, collision and so on. We have made extra explanations in the text. (please see lines 248-249)
**Reference:**
Bühl, J., Seifert, P., Myagkov, A., and Ansmann, A.: Measuring ice- and liquid-water properties in mixed-phase cloud layers at the Leipzig Cloudnet station, Atmos. Chem. Phys., 16, 10609–10620, https://doi.org/10.5194/acp-16-10609-2016, 2016.

In the discussions of figure 6e (and for 10e), it might not be clear to the reader how the calculations of INPC at different ice saturation ratios relate to the ICNC values. Does each saturation ratio relate to a different radius? A statement of which INPC and ICNC values are actually being compared for both part A and part B would make it clear to readers like me that are unfamiliar with the POLIPHON method.
**Response:** The saturation ratios ($S_i$) are endowed with 4 different values to simulate various degrees of heterogeneous nucleation, which allows us to infer the corresponding possible INPC values using parameterization U17-D. Note that part A and part B have been combined as a whole. A detailed explanation of computation and comparison has been added to the updated manuscript. Also, as suggested by reviewer #2, the Si used for pure dust (first case) and polluted/aged dust (second case) should be different, and the INPC for the polluted dust case is multiplied by a factor of 0.1 due to the weaker ice-nucleating efficiency of dust particles that have undergone the coating/aging process. Thus, we would like the reviewer to reevaluate the updated results and the related discussions. (please see lines 252-262, 309-324)

For part B starting at line 244, the message would be clearer if the comparison with the INPC came before putting it in the context of Ansmann et al (2019a), Cziczo et al (2013) and the interpretation of the dust acting as INP in the moist region. Additionally, if this is the moist region shown by the RH in figure 6d, this could be linked with a "as shown in figure 6d".
**Response:** We have made the necessary adjustments in the main text.

In the paragraph beginning on line 346 with "The overview of …. ", the authors might consider moving the sentences about the dust being uneven and the lack of INP resulting in homogeneous freezing near the start of the paragraph. They could then summarise the findings from part A and B for each case.
**Response:** As suggested by reviewer #2, homogeneous nucleation is rather unlikely as the presence of so many rather favorite dust INP particles as shown in figures 6 and 10, and the evidence for the depletion of INP is weak. Agreeing with this perspective, we have removed all the related argumentations regarding

homogeneous nucleation.

Related to the above point, a comment on why part B is shown to have lacking INP but is closest to the region of dust is perhaps warranted.
**Response:** As mentioned above, we have made major revisions to both cases by combining part A and part B as a whole as strongly suggested by reviewer #2. Now the results and discussions in section 3 have been significantly modified.

Line 375 states that cirrus would not form without these natural INP, but this conflicts with the establishment that homogeneous freezing does take appear to take place. Perhaps change to something similar to "would rarely form" or "there would be a much lower frequency of clouds".
**Response:** Thank you for pointing out this. Also, according to the comments from reviewer #3, we have modified the related expression. (please see lines 323-324)

The authors have done well to suggest plenty of future work based on this study.
**Response:** Thank you for your encouraging comments.

**Technical points**

There is some inconsistency between using the terms secondary ice nucleation and secondary ice production in lines 84 and 174.
**Response:** We have adopted "secondary ice nucleation" consistently throughout the revised manuscript.

Blue and purple colours in the profile plots (figures 6e and 10e) are quite hard to distinguish. These could be more contrasting colours.
**Response:** Considering that Part A and Part B are combined in each case as mentioned in previous responses, Figures 6e and 10e have been updated in the revised manuscript.

In figure captions, the authors could consider separating out figure titles from the a), b) descriptions. For example, the figure 2 caption could be "Dust-related conversion…" and then a) and b) to describe each plot. Also in figure 6 and 10.
**Response:** The related modifications have been done for the captions of figures 2, 6, and 10.

On line 164, what does a "complete system" mean here?
**Response:** The phrase "complete system" refers to the comprehensive functions of the HYSPLIT method, including analysis for air parcel trajectories as well as simulations involving transport, dispersion, chemical transformation, and deposition. A more detailed description of the HYSPLIT model has been added to the main text. (please see lines 171-173)

**Some typos:**

The abstract contains some use of the word "dominated" rather than dominant. This might be present in other places too.
Paragraph starting at line 164 contains several cases of HYSPIT instead of HYSPLIT.
Line 63, what is the occurrence rate of cirrus measured in?
Line 73: "Saseen et al 2003" should be Sassen
Line 75: should "Sassen et al 2001" be 2002?
Line 119: "MEERA" should be MERRA
Line 169: "stimulated" → simulated
**Response:** Thank you for pointing out the issue with the typos. For line 63, the occurrence rate of cirrus is measured in units of 0.3-0.4, i.e., 30%-40%. In response to all the mentioned issues, the necessary corrections have been made in the updated manuscript.

---

## Author Comment (AC4)

**Response to Reviewer #4**

**General Comments:**

The manuscript by Shen et al. presents an observational study of cirrus cloud formation near Midway Island in Pacific Ocean and the ice-nucleation properties of long-range transported desert dust. Authors have used observational data from Cloud-Aerosol Lidar with Orthogonal Polarization (CALIOP) and Cloud Profiling Radar (CPR), DARDAR and MERRA-2 datasets, and POLIPHON and HYSPLIT models for the data evaluation. As is, the manuscript addresses an important topic (especially for atmospheric and climate modeling communities) – the impact of long-range transport dust on the formation of cirrus clouds – but does not present an answer. Before the paper is published, some major concerns would have to be addressed.

My greatest concern with the manuscript is the lack of in-depth analysis – additional details (such as the nucleation mechanism, connection between the initial nucleation conditions and lab experiments) would improve the manuscript significantly. For this reason, these Major issues in the manuscript should be addressed.

**Response:** We sincerely appreciate the valuable feedback provided by the reviewer regarding our manuscript. Every comment has been thoughtfully taken into consideration, and necessary revisions have been made in the updated version. We consider that this revised manuscript has been largely improved. The main revisions are listed below:

(1) As strongly suggested by reviewer #2, here we decide to combine Part A and Part B and conduct the analysis by considering them as a whole, due to the uncertainty of DARDAR retrievals. The specific discussions have been largely updated now.

(2) As at least three reviewers suggest shifting this manuscript from 'Measurement Report' to 'Research Article', we would like to do so but also need to involve Handling Editor Prof. Krämer in the decision. In addition, considering the results have been largely rewritten by adding in-depth discussions, we also think it would be better to change it to 'Research Article'.

(3) As the cirrus clouds in the two cases have already formed for at least half an hour (as deduced from the vertical extent of the ice virga by assuming a falling velocity of 1 m/s), nucleated ice crystals may have undergone significant growth. Therefore, in the comparison between ICNC and INPC, we decide to mainly use the values of $n_{ice,25um}$ and $n_{ice,100um}$. The specific reason has been discussed in the text of the revised manuscript.

(4) As there is no evident indication of the depletion of dust INP, the possibility for occurring homogeneous nucleation is low. Therefore, we have removed all the discussions about the involvement of homogeneous nucleation in the cirrus clouds.

**Major issues:**

1. The idea and dataset chosen to show the nucleation and cloud formation are great, but the data analysis seems to be focused on an observation. It would be very interesting to see the atmospheric conditions data and the discussion of the initial heterogeneous ice nucleation (e.g., the transition from pure dust to pure cloud at 30.34/0.0 lat/lon on 2010-05-05 trajectory) to see the relation to the laboratory experiments (such as Koehler et al., Atmos. Chem. Phys., 10, 11955–11968, 2010).

**Response:** We are grateful for the reviewer's suggestion. Yes, as suggested, it would be better if we can connect the information from actual observations with that from the laboratory experiments. We have tried to do this in the revised manuscript as can be seen in the case study '**… In the case of polluted/coated dust, the ice-nucleating efficiency may possibly reduce to 10-20% of that for pure dust (Augustin-Bauditz et al., 2014; Wex et al., 2014); however, note that the coating-induced reduction in dust ice-nucleating efficiency is still not well-known quantificationally by the community. In addition, higher relative humidity is generally required for polluted/coated dust conditions (Koehler et al., 2010); Kärcher et al. (2022) used $S_i$ of around 1.35 for the activation threshold for polluted dust. Thus, if multiplying by a fact of 0.1, we can obtain a modified U17-D-derived INPCs of 310.1 L$^{-1}$; due to the quick growth of ice**

**crystals during their fall, we also reasonably use $n_{ice,25\ \mu m}$ and $n_{ice,100\ \mu m}$ for the comparison.'** (Please see lines 312-318)

**References:**

Koehler, K. A., Kreidenweis, S. M., DeMott, P. J., Petters, M. D., Prenni, A. J., and Möhler, O.: Laboratory investigations of the impact of mineral dust aerosol on cold cloud formation, Atmos. Chem. Phys., 10, 11955–11968, https://doi.org/10.5194/acp-10-11955-2010, 2010.

Augustin-Bauditz, S., Wex, H., Kanter, S., Ebert, M., Niedermeier, D., Stolz, F., Prager, A., and Stratmann, F.: The immersion mode ice nucleation behavior of mineral dusts: A comparison of different pure and surface modified dusts, Geophys. Res. Lett., 41, 7375–7382, https://doi.org/10.1002/2014GL061317, 2014.

Wex, H., DeMott, P. J., Tobo, Y., Hartmann, S., Rösch, M., Clauss, T., Tomsche, L., Niedermeier, D., and Stratmann, F.: Kaolinite particles as ice nuclei: learning from the use of different kaolinite samples and different coatings, Atmos. Chem. Phys., 14, 5529–5546, https://doi.org/10.5194/acp-14-5529-2014, 2014.

Kärcher, B., DeMott, P. J., Jensen, E. J., and Harrington, J. Y.: Studies on the competition between homogeneous and heterogeneous ice nucleation in cirrus formation, J. Geophys. Res.-Atmos., 127, e2021JD035805, https://doi.org/10.1029/2021JD035805, 2022.

2. The manuscript is presented as a Measurement Report, however it is not a new or original measurement, but rather a reanalysis of an old and public dataset. As a result, the manuscript does not provide substantial insight and conclusions. I would suggest the authors to perform and write-up an in-depth analysis of the processes based on their expertise and reconsider the manuscript as a research article.

**Response:** Thank you for the suggestion. Taking the reviewer's suggestion, we have removed 'Measurement report' in the title. Here we would also like to involve our Handling Editor Prof. Krämer to judge if it is justified to shift this manuscript from 'Measurement Report' to 'Research Article.' As the significant modifications have been made according to all the four reviewers' valuable comments, we hope that this manuscript may fulfill the requirements of a research article.

3. Another major shortcoming of the article is the Results section, where most of the section is focused on spelling out the results seen in figures with little interpretation. It is good to see that the summary of the results has been provided in Table 2, in the Discussion section, but I would expect it as part of Results.

**Response:** We have moved Table 2 to the results section and added a new subsection 3.3 to summarize the observational results of the two cases. Also, a new paragraph has been added in subsection 3.3. (Please see lines 337-352)

For these reasons, in my view, this work is not yet sufficient for publication and I would reconsider the manuscript after major revisions.

**Response:** After addressing the comments from all four reviewers, we have significantly modified the manuscript and hope that the updated work can be reconsidered by this reviewer.

**Minor issues:**

4. In Abstract, lines 24 and 25, the text says '[...] nucleation is dominated [...]' and '[...] nucleation can still be dominated [...]' while it should be 'dominant'.

**Response:** The word 'dominated' has been replaced with 'dominant'.

5. Line 70, 'concerned' should be 'considered'?

**Response:** The word 'concerned' has been replaced with 'considered'.

6. Mistypes and inconsistent labeling of the instrument and datasets: sections 2.1: MEERA-2; 2.5: HYSPIT.

**Response:** The title of section 2 has been updated to 'Instruments, Datasets, and Models'.

7. Missing satellite track, it should be added to the lon/lat maps. Vertical profile figures should have double latitude + longitude axis on abscissa (same as provided by CALIPSO).
**Response:** The figure 3, 4, 5, 7, 8, 9 have been updated in the revised manuscript.

8. The results section is riddled with 'Figure <n> shows [...]' sentences. I would suggest rephrasing them to 'As seen in Fig. <n>[...]', 'Based on data shown in Fig. <n>[...]', etc.
**Response:** We have revised the relevant sentences, reducing the use of 'Figure <n> shows' and employing alternative phrases.

9. Throughout the manuscript there are numerous grammatical errors (similar to the ones pointed out above) and it should be very carefully revised.
**Response:** The revised manuscript has been thoroughly checked to ensure there are no grammatical errors. Thank you for bringing this to our attention.

10. The authors have a significant number of self-citations that are present in addition to other previous works, e.g. (He et al., 2021b, 2022b), (Jing et al. 2023). I would suggest removing them where appropriate.
**Response:** We have reduced the occurrence of such self-citations.

11. The references (He et al., 2021a) and (He et al., 2022c) are not mentioned in the text at all. Please make sure that all unused references are removed.
**Response:** Thank you for pointing it out. They have been removed now.

12. The abstract provides a good summary, but is quite verbose and lengthy. If it is possible to shorten it without affecting quality, it would be excellent.
**Response:** We have rewritten the abstract according to the modifications in this round.

13. In Figure 8, the dust mass column density is for 2010-05-05, but the HYSPLIT trajectories span from 2008-04-23 to 2008-04-28. Is it a mistake in the caption, or was wrong dataset used for the figure?
**Response:** Thank you for pointing out this mistake. The correct date in Figure 8 should be 2008-04-27.